# A Closer Look at NTK Alignment: Linking Phase Transitions in Deep Image Regression

**Giuseppe Castiglione**
School of Engineering and Informatics
University of Sussex
g.castiglione@sussex.ac.uk

**Christopher Buckley**
School of Engineering and Informatics
University of Sussex
c.l.buckley@sussex.ac.uk

**Ivor Simpson**
School of Engineering and Informatics
University of Sussex
i.simpson@sussex.ac.uk

## Abstract

Deep neural networks trained with gradient descent exhibit varying rates of learning for different patterns. However, the complexity of fitting models to data makes direct elucidation of the dynamics of learned patterns challenging. To circumvent this, many works have opted to characterize phases of learning through summary statistics known as order parameters. In this work, we propose a unifying framework for constructing order parameters based on the Neural Tangent Kernel (NTK), in which the relationship with the data set is more transparent. In particular, we derive a local approximation of the NTK for a class of deep regression models (SIRENs) trained to reconstruct natural images. In so doing, we analytically connect three seemingly distinct phase transitions: the emergence of wave patterns in residuals (a novel observation), loss rate collapse, and NTK alignment. Our results provide a dynamical perspective on the observed biases of SIRENs, and deep image regression models more generally.

## 1 Introduction

Classical learning theory suggests that models with sufficient capacity - specifically, one whose parameters outnumber the training samples - tend to "memorise" individual examples rather than learn underlying patterns, leading to poor generalisation [1]. However, while Deep Neural Networks (DNNs) are typically over-parameterised, a growing body of research highlights the role of Gradient Descent (GD) in constraining their *effective* capacity [2, 3]. A recurring observation is that GD biases neural networks to prioritise learning simple patterns before more complex ones, resulting in distinct phases of learning [4, 5]. These phases are characterised by changes in the collective evolution of the network's weights, which can be quantified by statistics known as order parameters [6, 7, 8, 9]. Although numerous authors have independently proposed statistics to account for changes in convergence rate [10, 11] - and correspondingly, the memorisation [12] and over-fitting [13] of complex/noisy patterns - their interrelationships remain under-explored. More significantly, these existing approaches provide limited insight into the actual content being learned during each phase - and consequently, which patterns models systematically struggle to learn. Addressing these gaps is essential to developing a unified understanding of learning dynamics in DNNs.

A major obstacle in understanding this inductive bias lies in the inherent complexity of GD itself. While conceptually GD can be viewed as a function mapping from the dataset, hyperparameters, and initial weights to the final learned weights, in practice, the thousands of iterations through high-

dimensional parameter space obscure the relationship between order parameters and the underlying dataset characteristics. In recent years, the Neural Tangent Kernel (NTK) [14] has emerged as an alternative perspective on the dynamics of learning, recasting them in terms of the evolution of pointwise errors. Critically, in a phenomenon known as Neural Tangent Kernel Alignment (NTKA), the eigenspectra of the NTK undergo a sudden transition of their own, spontaneously aligning with the class structure of the dataset without direct supervision. NTKA has been widely documented and is suggested as a reason why real-world DNNs often outperform their infinite-width limit counterparts [15, 16, 17, 18, 19, 20]. However, despite repeated empirical demonstrations of NTKA, theoretical exploration of the phenomenon has been largely restricted to classification problems with toy models, such as two-layer neural networks [21, 22], and deep linear networks [22].

In this work, we move beyond these simple classification models and study NTKA in a considerably more complex setting: deep image regression using multi-layer SIRENs [23]. These Implicit Neural Representations (INRs) learn mappings from $\mathbb{R}^2 \to \mathbb{R}$, representing images as continuous functions, and find increasing application in tasks such as super-resolution. Despite the low input dimensionality, the depth and non-linear (sinusoidal) activations of these networks pose significant analytical challenges, exceeding the complexity of previously studied models. However, in addition to facilitating visualisation, this low-dimensionality permits us to leverage insights from computer vision to introspect the learning process. Our study is structured around three primary contributions:

1. We derive novel approximations for the local structure of the SIREN NTK, allowing us to approximate: the principal eigenvector (3.3); order parameters such as the minimum value of the Cosine NTK (3.4); and the correlation lengthscale (3.2). In so doing, we theoretically establish connections between the onset of NTKA and other dynamical phase transitions.

2. We identify a novel learning phase in deep image regression, characterized by the appearance of diffusion-like wavecrests in the residuals, and relate this behaviour to the evolution of the NTK.

3. We experimentally verify that the critical points for these different phase transitions cluster in time. We also empirically investigate the impact of image complexity and SIREN hyperparameters on the occurrence and timing of phase transitions, and provide evidence that NTK alignment in image regression tasks occurs in response to difficulties in modelling edges.

## 2 Preliminaries

In this work, we consider 2D grayscale images, where pixel coordinates and their intensity form a dataset $\mathcal{D}$ of $N$ samples indexed with $i$, $(x_i, I(x_i))$, where $x_i \in \mathbb{R}^2$ and $I : \mathbb{R}^2 \mapsto \mathbb{R}$. On this dataset, we fit SIREN models $f(x; \theta)$ of depth $N_l$, defined recursively by: $h^{(0)} = x$; $h^{(l)} = \sin \omega_0 (W^{(l)} h^{(l-1)} + b^{(l)})$; $f(x; \theta) = W^{(N_l)} h^{(N_l-1)} + b^{(N_l)}$. Here $h^{(l)}$ denotes the output of the $l$-th layer, $\theta = \{W^{(l)}, b^{(l)} | l = 1, \ldots, N_l\}$ is the set of learnable parameters, and $\omega_0$ is a bandwidth hyperparameter. $\omega_0$ is generally chosen to ensure the $\sin$ function spans multiple periods (and thus frequencies) over the inputs. In the continuum limit, we assume the data is distributed uniformly $P_{data}(x) = \text{Vol}(\mathcal{D})^{-1}$. We identify two fields: the local residual field $r(x; \theta(t)) = I(x) - f(x; \theta(t))$, and gradient field $\nabla_\theta f(x; \theta(t))$. Dynamics are induced by gradient flow $\dot{\theta} = -\nabla_\theta L$ on the mean square error: $L(\theta) = \frac{1}{2\text{Vol}(\mathcal{D})} \int dx \, r(x; \theta)^2$. Through the chain rule, the residuals evolve as follows:

$$\dot{r}(x; \theta(t)) = \nabla_\theta r(x; \theta(t)) \cdot \dot{\theta} \tag{1}$$

$$= -\frac{1}{\text{Vol}(\mathcal{D})} \int dx' \, r(x') \nabla_\theta r(x; \theta(t)) \cdot \nabla_\theta r(x'; \theta(t)) \tag{2}$$

$$= -\int dx' \, r(x') \underbrace{\left( \frac{1}{\text{Vol}(\mathcal{D})} \nabla_\theta f(x; \theta(t)) \cdot \nabla_\theta f(x'; \theta(t)) \right)}_{K_{NTK}(x, x'; \theta(t))} \tag{3}$$

In the last line, we defined the NTK. Equation 3 is a linear dynamical system with a time-varying kernel. The eigenvectors $v_k(x, t)$ represent distinct normal modes of the dataset, each learning at a rate governed by its associated eigenvalue $\lambda_k(t)$. This framework formalizes the intuitive notion that neural networks learn different patterns at different speeds.

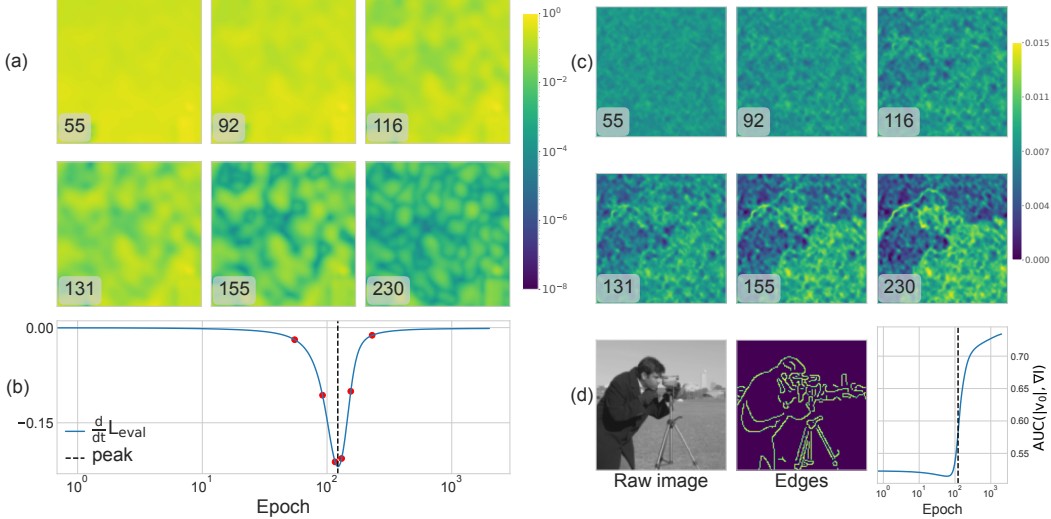

Figure 1: **A Single Phase Transition Through Three Lenses**: (a) The magnitude of the residuals over the training process. Near the critical point we see the formation of wavecrests. (b) Evolution of evaluation loss rate during training, which reaches a peak at the critical point. (c) Evolution of the principle eigenvector of the NTK, which reveals a sudden shift from disorder to structure. (d) Quantification of NTKA in terms of alignment between edges and the principal eigenvector.

Finally, for notational brevity, we will drop the explicit dependence on $\theta$, and write $x' = x + u$. We also define a kernel closely related to the NTK, the Cos NTK:

$$C_{NTK}(x, x+u) = \frac{1}{\text{Vol}(\mathcal{D})} \frac{\nabla_\theta f(x) \cdot \nabla_\theta f(x+u)}{||\nabla_\theta f(x)|| \, ||\nabla_\theta f(x+u)||} \tag{4}$$

## 3 Deriving Order Parameters from the NTK

We illustrate the different phases of learning in Figure 1; we train a (five-layer, 256-unit wide, $\omega_0 = 60$) SIREN model on a $128 \times 128$ grayscale image using full-batch GD (learning rate=$10^{-3}$). We evaluate the model on super-resolution at $256 \times 256$. We examine the learning dynamics through three different lenses, each revealing a sudden shift These shifts are quantitatively identified using statistics, known as order parameters. We demonstrate below how order paramters for each transition may be related to a common set of features, which control the local NTK structure. The three lenses are as follows:

- **Spatial Distribution of Residuals**: Early in training, the loss decreases uniformly over the dataset (Drift Phase). However, at a critical point, we observe the formation of "wave-crests" corresponding to regions of low-loss, which propagate across the dataset (Diffusion Phase). To the best of our knowledge, we are the first to report this behaviour in SIREN models. We attribute this behaviour (in sec. 3.1) to changes in the equal-time correlation functions of the gradient field $\nabla_\theta f(x)$, whose parameters we derive in Section 3.2.

- **Principal Eigenvectors of the NTK**: The principal eigenvector $v_0$ is initially static and appears highly-disordered (Disordered Phase). However, at a critical point, $v_0$ experiences a brief, sudden shift, in which it aligns with the edges of the image (Aligned Phase). Although NTKA has previously been studied in the context of classification problems [24, 25, 26, 27], there are additional subtleties to consider for a regression task such as INR training. To this end, we introduce a metric, AUC($|v_0|, \nabla I$) in Section 3.3 to identify when alignment occurs. We also derive an approximation of $v_0$ based on the local structure of the NTK, as outlined in Sections 3.1 and 3.2.

- **Training Curve Analysis**: There is a rapid shift in the slope of the training curve, which we call the loss rate $\dot{L}$. Learning is initially fast (high $\dot{L}$), but after a critical point, slows abruptly (low $\dot{L}$). Several works have studied this transition using order parameters, but in this work, we focus on

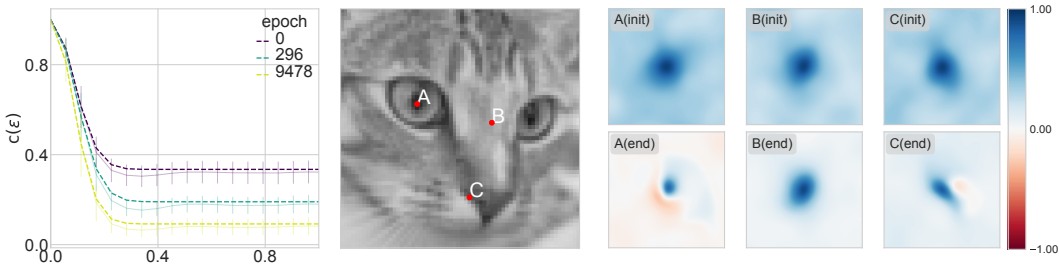

Figure 2: **Spatiotemporal Evolution of the Cosine NTK** ($C_{NTK}$). Left: Global correlation function of the $C_{NTK}$ at different epochs. Dashed lines show fitted Gaussian approximation from equation equation 6, and error bars show variance across datasets. Right: Visualization of the $C_{NTK}$ around three points $x \in \{A, B, C\}$ for small separations, at the beginning and end of training.

the concept of gradient confusion, as described in [13], [11], [12]. In Section 3.4, we derive an approximation of this parameter based on the local structure of the NTK outlined in Section 3.2.

### 3.1 Correlation Functions and the Onset of Diffusion

The form of equation 3 is reminiscent of the linear response functions in statistical field theory [28, 29]: to find the rate of change of the residual field at a point $x$, the kernel $K$ aggregates information about the residual at points $x + u$. To quantify the range of these interactions, we examine the local, equal-time correlation functions for the gradients $\nabla_\theta f(x)$ separated by a distance $\epsilon$:

$$k(x, \epsilon) = \mathbb{E}_\phi\big[\nabla_\theta f(x) \cdot \nabla_\theta f(x + \epsilon\hat{e}_\phi)\big] = \mathbb{E}_\phi\big[K_{NTK}(x, x + \epsilon\hat{e}_\phi)\big] \quad (5)$$

Here, $\hat{e}_\phi$ denotes a unit vector in direction $\phi$. Similarly, the global, equal-time correlation function is given by $k(\epsilon) = \mathbb{E}_x\big[k(x, \epsilon)\big]$. We may define similar quantities for the $C_{NTk}$, which we denote by $c(x, \epsilon)$ and $c(\epsilon)$. We expect the range of these interactions to be short, as INRs are often carefully designed to ensure a diagonally dominant NTK[30, 31, 32]. To verify this, we group pairs of datapoints based on their distance, and then compute the mean $C_{NTk}$ value. An example of this empirical correlation function is shown in Figure 2. The observed structure motivates the following proposition, which we examine qualitatively in Figure 12 and validate numerically in Appendix F.1.

**Proposition 3.1.** *For SIREN models, the equal-time correlation functions of the NTK are well-approximated by Gaussians of the form:*

$$c(\epsilon) \approx (1 - c_\infty)e^{-\epsilon^2/2\xi_{corr}^2} + c_\infty, \quad (6)$$

$$k(x, \epsilon) \approx ||\nabla_\theta f(x)||^2\big[(1 - c_\infty(x))e^{-\epsilon^2/2\xi(x)^2} + c_\infty(x)\big]. \quad (7)$$

Our approximation introduces two important order parameters: the first, the correlation length-scale $\xi_{corr}$, controls the rate at which correlations decay with distance, defining the range of interactions. The second, the asymptotic value $c_\infty$, describes the interactions between points at separations $\epsilon$ much greater than $\xi$. Suspending a consideration of anisotropic effects, the above correlation functions are consistent with the following form for the full NTK:

**Proposition 3.2.** *Gaussian Approximation of the SIREN NTK. The NTK may be be approximated as a Gaussian kernel with spatially-varying amplitude $||\nabla_\theta f(x)||^2(1 - c_\infty(x))$, bandwidth $\xi(x)$, and asymptotic value $||\nabla_\theta f(x)||^2 c_\infty(x)$:*

$$K_{Gauss}(x, x + u) \approx ||\nabla_\theta f(x)||^2(1 - c_\infty(x)) \exp(-||u||^2/\xi^2(x)) + ||\nabla_\theta f(x)||^2 c_\infty(x) \quad (8)$$

Dynamically, we see from the left of Figure 2 that both $\xi$ and $c_\infty$ evolve during training, and we shall demonstrate that changes in these values account for the onset of diffusion.

**Theorem 3.1.** *Diffusive Evolution of the Residuals: Let the mean residual be denoted by $\mu_r \equiv \mathbb{E}_x[r]$, and let $K_\infty$ denote the mean contribution to the $K_{NTK}$ at large distances $||u|| \gg \xi(x)$. Then, as $\mu_r K_\infty \to 0$, assuming $K_{NTK} \approx K_{Gauss}$, the residuals approximately evolve under the following diffusion equation:*

$$\frac{d}{dt}r(x, t) \approx -2\pi\xi^2(x)||\nabla_\theta f(x)||^2 r(x, t) - \pi\xi^4(x)||\nabla_\theta f(x)||^2 \Delta_x^2 r(x, t) \quad (9)$$

*Proof Sketch.* (Full details in Appendix A.2). As $\mu_r K_\infty \to 0$, local interactions dominate the background in determining the evolution of $r(x)$ in equation 3. Thus we may approximate:

$$\frac{dr}{dt} \approx - \int dx \, r(x+u)K(x, x+u) \approx \int dx \, r(x+u)||\nabla_\theta f(x)||^2 \exp(-||u||^2/\xi^2(x)) \quad (10)$$

The exponentially-decaying NTK will suppress all contributions to $\dot{r}(x)$ from residuals $r(x+u)$ for which $||u|| \gg \xi(x)$. When $\xi(x)$ is small, we may perform a Taylor expansion around the point $x$:

$$r(x+u; \theta) \approx r(x; \theta) + u^\top \nabla_x r(x; \theta) + \frac{1}{2}u^\top \nabla_x^2 r(x; \theta)u \quad (11)$$

Inserting this into equation 10, the full integral may be solved using Gaussian integration. We obtain:

$$\frac{d}{dt}r(x; \theta) = -2\pi\xi^2(x)||\nabla_\theta f(x)||^2 r(x) - \pi\xi^4(x)||\nabla_\theta f(x)||^2 \Delta_x^2 r, \quad (12)$$

which resembles a standard diffusion equation $\qquad\qquad\qquad\qquad\qquad\qquad\qquad$ □.

## 3.2 Beyond the Isotropic Gaussian Approximation

Though the Gaussian approximation of the NTK can account for the appearance of the diffusion wavecrests, it fails to capture other empirical properties. Namely, whereas the true NTK is anisotropic (see Figure 2) and can contain negative entries, the Gaussian kernel is isotropic, and satisfies $K_{Gauss}(x, x+u) > 0$ for all $u$. To overcome these limitations, we present the following refinement:

**Theorem 3.2.** *Cauchy Approximation of the SIREN NTK: For small separations $u$, the Cosine NTK locally takes the form of a Cauchy Distribution with structure parameters $a_x, D_x, H_x$:*

$$C_{NTK}(x, x+u) \approx \frac{2a_x^2 + u^\top D_x}{2a_x^2 + u^\top D_x + u^\top H_x u}, \quad (13)$$

*These parameters are obtained from the model gradients as follows:*

$$a_x = ||\nabla_\theta f(x)||; \ D_x = \nabla_x ||\nabla_\theta f(x)||^2; \ H_x = (\nabla_x \nabla_\theta f(x))(\nabla_x \nabla_\theta f(x))^\top \quad (14)$$

*Proof Sketch.* (Full details in Appendix A.3). Via the Law of Cosines, the $C_{NTK}$ satisfies:

$$C_{NTK}(x, x+u) = \frac{||\nabla_\theta f(x)||^2 + ||\nabla_\theta f(x+u)||^2 - ||\nabla_\theta f(x+u) - \nabla_\theta f(x)||^2}{2||\nabla_\theta f(x)|| \, ||\nabla_\theta f(x+u)||} \quad (15)$$

The result then follows by Taylor expanding the numerator and denominator to second order in $u$. □

A benefit of this new approximation is that it can be used to predict the correlation length-scale:

**Corollary 3.2.1.** *The correlation length-scale for the NTK about a point $x$ may be constructed from its local structure parameters $a_x, D_x, H_x$ and asymptotic value $c_\infty(x)$ as follows:*

$$\xi^2(x) \approx 2\left(\frac{1-c_\infty(x)}{1+c_\infty(x)}\right)\frac{a_x^2}{\sqrt{\det H_x}} + \frac{1}{4}\left(\frac{1-c_\infty(x)}{1+c_\infty(x)}\right)^2 \frac{D_x^\top H_x^{-1} D_x}{\sqrt{\det H_x}} \quad (16)$$

*Proof Sketch.* (Full details in Appendix A.4). Note that the levels sets of equation 13 correspond to ellipses. For a given value $c$, the area of the level set can be shown to be:

$$A_{ellipse}(x; c) = 2\pi\left(\frac{1-c}{c}\right)\frac{a_x^2}{\sqrt{\det H_x}} + \frac{\pi}{4}\left(\frac{1-c}{c}\right)^2 \frac{D_x^\top H_x^{-1} D_x}{\sqrt{\det H_x}} \quad (17)$$

The correlation lengthscale is then approximated as $\xi(x) \approx \sqrt{A_{ellipse}(x, c)/\pi}$, where we choose $c = 1/2 + c_\infty/2$ to account for the asymptotic value of the $C_{NTK}$. □

### 3.3 Order Parameters for the Onset of NTK Alignment

In the classification problems typically studied in the NTKA literature, the principle eigenvector $v_0(x)$ is seen to learn class-separating boundaries [24, 25]. Similarly, for our 2D image reconstruction task, we see the NTK learns information about the distribution of edges in the image (Figure 3). To quantify this alignment, we use a Canny Edge Detector [33] to estimate connected image edges. We then quantify the utility of $|v_0(x)|$ in predicting edges in terms of average recall, as measured by the area under the Receiver Operating Characteristic Curve (ROC AUC). We denote this measure $\text{AUC}(|v_0|, \nabla I)$, and it has the advantage of being insensitive to monotonic transformations of $|v_0|$. This invariance is beneficial in two respects: (1) as a reliability measure for a binary predictor (edges), it obviates the need to specify a threshold, facilitating comparisons across datasets (see Section 4.1); and (2) empirically, it saturates during training, facilitating the the identification of a critical point.

Another hallmark of NTKA is early anisotropic growth of the NTK spectrum [25], as the NTK becomes stretched along a small number of directions correlated with the task. This is especially the case for the principal eigenvalue $\lambda_0$, which grows orders of magnitude larger than the next leading eigenvalue. In Section 4.1, we will demonstrate empirically that this is also true for INRs.

The divergence of $\lambda_0$ enables a particularly simple approximation of the princpal eigenvector $v_0$:

**Corollary 3.2.2.** *The principal eigenvector $v_0(x)$ of the NTK admits the following approximation in terms of the local asymptotic value $c_\infty(x)$ and the local correlation lengthscale $\xi(x)$:*

$$v_0(x) \approx a_x^2 \left[ c_\infty(x)\text{Vol}(\mathcal{D}) + 2\pi\xi^2(x)(1 - c_\infty(x)) \right] \tag{18}$$

*Proof.* Because the principal eigenvalue is so dominant, $K_{NTK}$ becomes effectively low-rank, and so power iterations converge quickly. Thus, choosing a vector of ones $v = 1$ as our initial vector, we expect $K1/1^\top 1$ to have strong cosine alignment with the principal eigenvalue. In the continuum limit, this is simply given by:

$$K1/N \to \mathbb{E}_u[K(x, x + u)] = \mathbb{E}_\epsilon[\mathbb{E}_u[K(x, x + u)| \ ||u|| = \epsilon]] \tag{19}$$

$$= \int_0^{\epsilon_{max}} d\epsilon \ k(x, \epsilon)P(x, \epsilon) \tag{20}$$

Here, $P(x, \epsilon)$ denotes the density of points that are located a distance $\epsilon$ from the point $x$, and $\epsilon_{max}$ is an upper bound on the distance that we assume is much greater than $\xi_{corr}$. Close to this $x^1$, $P(x, \epsilon)$ grows like $2\pi\epsilon$. Thus, leveraging equations 7 and 14, we have:

$$v_0(x) \approx 2\pi a_x^2 \int_0^{\epsilon_{max}} d\epsilon \ \epsilon \left[ c_\infty(x) + (1 - c_\infty(x))e^{-\epsilon^2/2\xi^2(x)} \right] \tag{21}$$

$$= 2\pi a_x^2 \left[ c_\infty(x)\epsilon_{max}^2 + \xi^2(x)(1 - c_\infty(x))(1 - e^{-\epsilon_{max}^2/2\xi^2(x)}) \right] \tag{22}$$

$$\approx a_x^2 \left[ c_\infty(x)\text{Vol}(\mathcal{D}) + 2\pi\xi^2(x)(1 - c_\infty(x)) \right] \qquad \square$$

We evaluate the fidelity of this approximation in Appendix F. As we approach the phase transition, the asymptotic values tend towards 0, and the second term dominates. Considering the approximation for the correlation length-scale $\xi$ in Corollary 3.2.1, we note that $v_0(x)$ grows as $\mathcal{O}(||\nabla_\theta f(x)||^4)$. This implies particular sensitivity to pixels in regions with substantial high-frequency information, such as edges and corners. As natural images tend to be piecewise smooth, pixels on boundaries have the strongest spatial gradients, and are therefore the greatest source of information, being poorly compressible due to the lack of smoothness, and accordingly disagreement in parameter gradients. Given the inability of models to accurately describe sharp discontinuities these edge pixels are considered influential datapoints, which accounts for their prominence within the principal eigenvector. We discuss other parallels between the moments of the NTK and traditional corners detection algorithms in Appendix E. In particular, we introduce another order parameter, termed **MAG-Ma** (**M**agnitude of the **A**verage **G**radient of the Log Gradient-Field **Ma**gnitudes), to monitor the breakdown of stationarity (ie local translation invariance) of the NTK. It is obtained as $||\mathbb{E}_x[D_x/a_x^2]||^2$.

---

[1]The true form of $P(x, \epsilon)$ is complicated and varies from point to point, due to edge effects. However, these effects are suppressed as $P(x, \epsilon)$ only appears when multiplied the Gaussian $k_x$.

### 3.4 Order Parameters for the Loss Rate Collapse

In [13], [11], [12], and related works, the authors examine the role of gradient alignment statistics in determining the speed of learning under stochastic gradient descent. They note the emergence of negative alignments between batches correlates with a reduction in learning speed. Intuitively, when sample gradients become negatively aligned, the sum of the gradients approaches zero, resulting in a diminished learning signal. The minimum alignment is simply the minimum value of the $C_{NTK}$, which we may obtain explicitly from Theorem 3.2 as follows:

**Corollary 3.2.3.** *The minimum value of the $C_{NTK}$ admits the following approximation in terms of the local structure parameters $a_x, D_x, H_x$:*

$$\min_u C_{NTK}(x, x + u) = \frac{D_x^\top H_x^{-1} D_x}{D_x^\top H_x^{-1} D_x - 8a_x^2} \tag{23}$$

*Proof Sketch.* (Full details in Appendix A.5). Setting $\partial_u C_{NTK}(x, x + u) = 0$ yields two solutions: $u = 0$, corresponding to the maximum (1), and another corresponding to the minimum. $\square$

The $\min C_{NTK}$ is then simply the minimum of 23 across the whole dataset.

## 4 Experimental Results

**Setup**: We fit SIREN models to a set of thirty $64 \times 64$ downsampled images and evaluate the MSE $L_{eval}$ on a super-resolution task (at $256 \times 256$). We used five random seeds and also varied the width, depth and bandwidth $\omega_0$ (ranges are given in Appendix B.2). We compute the eigenspectra of the NTK using Randomized SVD [34]. In addition to the order parameters described in Section 3, we examine three NTK-based order parameters from the literature: (1) The principal eigenvalue $\lambda_0$ of the NTK, which diverges at the critical point; (2) The variance of the gradients $\sigma_\theta^2$, which peak during the Fast-Slow learning phases [10], and which may be connected (see Appendix A.6) to the trace of the NTK; (3) The Centred Kernel Alignment (CKA) between the NTK and a task kernel $K_Y$. For INR regression, we use $K_Y(x, x + u) = \exp(-50||I(x) - I(x + u)||^2)$. The similarity between kernels is measured using the normalized Hilbert-Schmidt Information Criterion (HSIC), as in [25, 26, 27]. Full experimental details may be found in Appendix B.2.

### 4.1 Examining the Distribution of Critical Points

**Critical points cluster around run-specific times**: The left-hand side of Figure 3 illustrates our procedure for identifying critical points in a given trial. We use a simple peak detector to identify the region of interest for the loss rate $\dot{L}_{eval}$ and the gradient variance $\sigma_\theta$, using the FWHM to define a confidence region. For the $\min C_{NTK}$, we look for zero-crossings, with a confidence region constructed from the cumulative variance. For every other order parameter, we fit a sigmoid, where the inflection point marks the critical point, and the slope defines the confidence region (full details in Appendix B.2). The right side of Figure 3 demonstrates how frequently these confidence regions overlap across our experimental sweep[2]. Remarkably, the phase transitions described by the order parameters - despite being derived to measure different phenomenon in the literature - consistently occur at the same time during training.

**Hyperparameters alter the timing of run-specific transitions**: in Table 1 we observe that both depth and bandwidth $\omega_0$ have critical roles in controlling the shift in the loss rate $\dot{L}_{eval}$. Generally, increasing depth and decreasing $\omega_0$ result in earlier transition times $t_{crit}$. However, these changes have opposite effects on the model performance: for fixed $\omega_0$, deeper models to converge to better (lower $L_{eval}$) solutions faster. However, it seems that lower values of $\omega_0$ cause models to converge prematurely. This may be deduced by studying the correlation between the final residuals (details in Appendix B.2). For equivalent depth (and therefore, equivalent traditional capacity), models with lower $\omega_0$ exhibit more correlations in the residuals. This is indicative of remaining structure in the residuals, and can be interpreted as evidence of under-fitting.

---

[2]In computing the coincidence matrix, we exclude trials where the detection of the critical point was unreliable. In Appendix C, we comment on how image properties impact the detection rate.

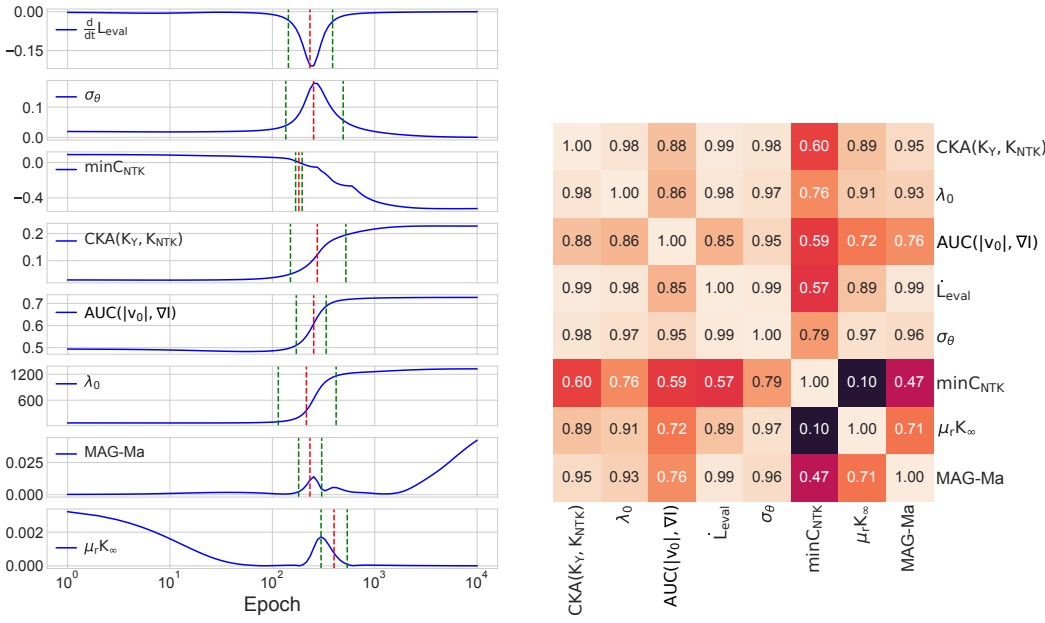

Figure 3: **Alignment of Order Parameters**. Left: Order parameter evolution and critical points during training of a SIREN model on the astro image. The red vertical lines denote the location of the critical points, and the green vertical lines denote confidence regions. Right: Heatmap showing the frequency of intersections between the confidence regions. Additional figures in Appendix G.1.

Table 1: **Variation in model performance with hyperparamters**: Comparing the dependence of transition times ($t_{crit}$), super-resolution performance ($\log_{10} L_{\text{eval}}$), and resdual correlation on depth and bandwidth. Also featuring expected correlation lengthscale ($\mathbb{E}[\xi_{corr}(t)]$) and correlation between $\log ||\nabla_\theta f(x; \theta)||$ and $||\nabla_x I(x)||$. Values are averaged over the same sweep defined in Section 4.1.

| $\frac{depth}{\omega_0}$ | $\mathbb{E}[\xi_{corr}(t)]$ | AUC($|v_0|, I$) | Grad. Corr. | $\log_{10} t_{crit}$ | $\log_{10} L_{\text{eval}}$ | Res. Corr. |
|---|---|---|---|---|---|---|
| 3/90 | $0.04 \pm 0.00$ | $0.59 \pm 0.05$ | $0.06 \pm 0.10$ | $2.83 \pm 0.19$ | $-2.01 \pm 0.31$ | $0.35 \pm 0.06$ |
| 3/60 | $0.06 \pm 0.00$ | $0.60 \pm 0.05$ | $0.16 \pm 0.11$ | $2.84 \pm 0.12$ | $-2.00 \pm 0.32$ | $0.40 \pm 0.07$ |
| 3/30 | $0.11 \pm 0.00$ | $0.60 \pm 0.05$ | $0.27 \pm 0.08$ | $2.44 \pm 0.25$ | $-1.92 \pm 0.30$ | $0.44 \pm 0.07$ |
| 3/15 | $0.18 \pm 0.01$ | $0.60 \pm 0.05$ | $0.26 \pm 0.07$ | $2.03 \pm 0.34$ | $-1.83 \pm 0.29$ | $0.48 \pm 0.07$ |
| 4/90 | $0.04 \pm 0.00$ | $0.67 \pm 0.07$ | $0.23 \pm 0.13$ | $2.66 \pm 0.17$ | $-2.02 \pm 0.32$ | $0.35 \pm 0.06$ |
| 4/60 | $0.06 \pm 0.00$ | $0.69 \pm 0.07$ | $0.29 \pm 0.12$ | $2.61 \pm 0.16$ | $-2.04 \pm 0.33$ | $0.39 \pm 0.07$ |
| 4/30 | $0.10 \pm 0.00$ | $0.71 \pm 0.08$ | $0.41 \pm 0.10$ | $2.22 \pm 0.27$ | $-1.99 \pm 0.31$ | $0.41 \pm 0.07$ |
| 4/15 | $0.16 \pm 0.01$ | $0.68 \pm 0.09$ | $0.55 \pm 0.08$ | $1.87 \pm 0.35$ | $-1.94 \pm 0.30$ | $0.43 \pm 0.07$ |
| 5/90 | $0.03 \pm 0.00$ | $0.68 \pm 0.07$ | $0.24 \pm 0.14$ | $2.54 \pm 0.18$ | $-2.01 \pm 0.32$ | $0.34 \pm 0.06$ |
| 5/60 | $0.05 \pm 0.00$ | $0.71 \pm 0.07$ | $0.30 \pm 0.13$ | $2.42 \pm 0.17$ | $-2.05 \pm 0.33$ | $0.39 \pm 0.07$ |
| 5/30 | $0.09 \pm 0.00$ | $0.73 \pm 0.08$ | $0.39 \pm 0.12$ | $2.15 \pm 0.12$ | $-2.02 \pm 0.32$ | $0.40 \pm 0.07$ |
| 5/15 | $0.15 \pm 0.01$ | $0.72 \pm 0.08$ | $0.52 \pm 0.09$ | $1.85 \pm 0.27$ | $-1.98 \pm 0.31$ | $0.41 \pm 0.07$ |

## 4.2 Influence of Hyperparameters on Edge Alignment

In the previous section, for fixed depth, we demonstrated that lower $\omega_0$ correlates with (1) earlier phase transitions, (2) higher validation loss, and (3) correlated residuals. Together, these observations suggest that models with lower $\omega_0$ converge prematurely, underutilizing their capacity. A natural question arises: which patterns do these models struggle to capture? Given the observed concurrence of loss rate collapse and NTK alignment, we analyse the NTK eigenspectrum to gain some insight.

In Figure 4, we train four SIRENs on the sax dataset with $(\omega_0, \text{depth}) \in \{15, 60\} \times \{3, 5\}$. We visualise both the log magnitudes of the parameter gradients, $\log ||\nabla_\theta(x) f||^2$, and the principal eigenvector, $v_0(x)$, at the end of training. Additional Figures may be found in Appendix C.2

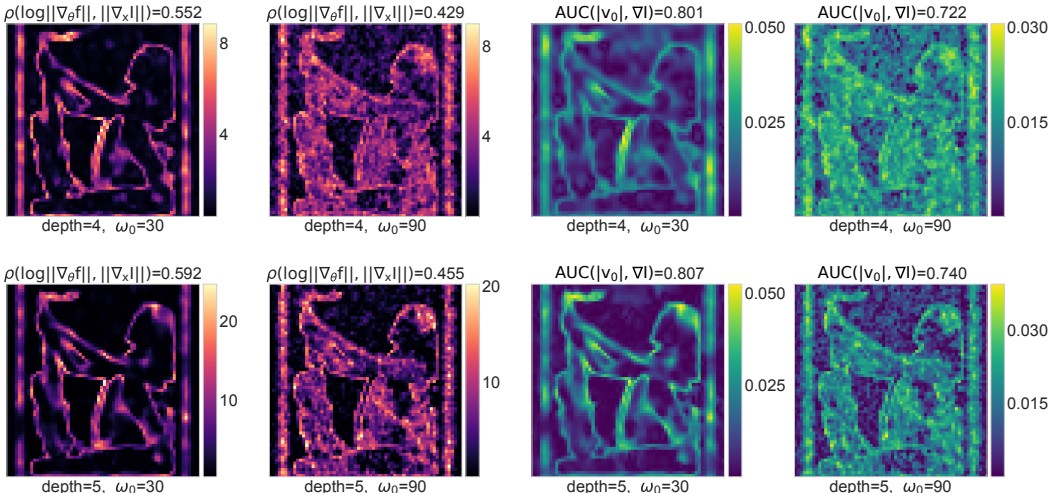

Figure 4: **Effect of Hyperparameters on Edge Alignment**: Left (magma colormap): The norm of the parameter gradients $||\nabla_\theta f(x)||$ for $(\omega_0, \text{depth}) \in \{15, 60\} \times \{3, 5\}$, labelled with the Pearson correlation $\rho$ between the log of the norm and the spatial gradient $\nabla_x I$ of the target image. Right (viridis colormap): visualizing the principal eigenvector $v_0$ of the NTK for the same models, labelled with the edge alignment score $\text{AUC}(|v_0|, \nabla I)$. More images in Appendix C.2

Generally, we observe that for low $\omega_0$, $\log ||\nabla_\theta f(x; \theta)||^2$ swells and concentrates near image edges, becoming sparser, and more correlated with the spatial gradient magnitudes $||\nabla_x I(x)||$ (aggregated statistics may be seen in Table 1). This edge prominence in $v_0$ matches expectations, as per Corollary 3.2.2, $v_0(x) \sim \mathcal{O}(||\nabla_\theta f(x)||^4)$. Overall, while edge alignment is seen across most settings, it is especially prominent for deeper models with lower values of $\omega_0$. This indicates prioritization of these patterns by the NTK, and correspondingly, the patterns the model is most invested in.

## 5 Discussion

To explain the impact of $\omega_0$, in Table 1 we track the expected correlation lengthscale $\mathbb{E}[\xi_{corr}(t)]$ over the course of training. In Figure 17 of the Supplementary Materials, we also examine the variance in $\xi_{corr}(0)$ with $\omega_0$ in all models. Consistently, we observe that lower $\omega_0$ is associated with larger values of the correlation lengthscale. This implies that the NTK integrates information across larger neighborhoods, implicitly averaging over high-frequency features. Consequently, we expect these SIRENs to struggle when modeling high-frequency patterns, which is consistent with other observations in the literature [32]. Following the discussion in Section 3.3, we expect that the difficulty of modeling edges is responsible for the swelling of the parameter gradients $||\nabla_\theta f(x)||^2$.

Intriguingly, we observe that the principal eigenvector becomes sparser as we increase depth, leading to stronger edge alignment, as seen in Table 1. Yet, this sparsification is associated with completely different generalization behavior: models achieve lower validation loss, with less correlated residuals, as we increase depth. We hypothesise a different mechanism underlies this sparsification in comparison with $\omega_0$. Figure 4 demonstrates increasing $\omega_0$ increases the sensitivity of the gradient magnitudes (and hence the principal eigenvector) to noise in the images. For DNNs, gradient magnitudes decompose into a sum across layers, namely $||\nabla_\theta f||^2 = \sum_{l=1}^{\text{depth}} ||\nabla_{\theta^{(l)}} f||^2$. In effect, preference is given to points which are consistently confusing across layers, thus mitigating the effects of noise.

## 6 Related Work

**Fast and Slow Phases of Neural Network Training**: The literature highlights a dynamical phase transition in DNN training between fast to slow learning regimes [10]. The initial fast phase is characterized by large gradient norms and low fluctuations, yielding rapid loss reduction via broad agreement across examples. It is followed by a slow phase in which gradient fluctuations dominate

and progress decelerates. This transition can be tracked by order parameters including gradient signal-to-noise [10], gradient confusion, [13, 11, 12], and correlation lengthscales [13]. Furthermore, it reflects a move from learning simple, shared patterns to fitting complex, idiosyncratic ones [4], and thus offers a window into feature learning. Related work studies the representation dynamics in ReLU networks [35]; here, we study them via the NTK in SIREN models.

**Neural Tangent Kernels for Implicit Neural Representations**: Previous research has investigated the inductive biases of INRs using the Neural Tangent Kernel (NTK), focusing on aspects such as spectral properties [36] and dependencies on uniformly sampled data [30]. Furthermore, studies by [37] and [38] have analyzed the eigenfunctions of the empirical NTK to elucidate the approximation capabilities of INRs. These investigations, however, primarily examine static properties of the NTK at initialization, which do not account for feature learning dynamics. This is known to be a poor approximation [39]. In contrast, our work concentrates on the evolution of the NTK, aiming to deepen our understanding of how INRs learn to model images.

**Neural Tangent Kernel Alignment** In practical settings, recent studies have shown that during training, the NTK dynamically aligns with a limited number of task-relevant directions [40, 41, 24, 21, 25, 22, 26, 27]. Concurrently, at the eigenfunction level, the modes increasingly reflect salient features of the dataset, such as class-separating boundaries [24, 25], and Fourier frequencies [25]. The widespread occurrence and influence of kernel alignment suggest its critical role in DNN feature learning, contributing to the superior performance of DNNs over models based on infinite-width NTKs [26]. Direct optimisation of alignment measures has even been suggested as one way to enhance the convergence of GD and generalization of models[42, 43]. That said, theoretical investigation into spontaneous NTKA often focus on shallow networks [21, 22], toy models [26, 25], and deep linear networks [22]. In contrast, the INRs we study are deep (3-6 layers), nonlinear models that see frequent use in Computer Vision problems.

# 7   Conclusion

We have developed new formulations that leverage the NTK to characterise the dynamics of feature learning in deep image regression models (SIRENs). By analytically deriving approximations for the local structure of SIREN NTKs - using Gaussian and Cauchy distributions - we were able to obtain approximate expressions for the correlation lengthscale, the minimum value of the $C_{NTK}$, and the principal eigenvector. We related these expressions to order parameters for three phase transitions identified in different dynamical perspectives on learning: the appearance of diffusion wave-crests in residual evolution (first identified in this paper); the collapse of the loss rate; the onset of NTK alignment. We argued, based on these derivations and empirical demonstrations that critical points cluster in time, that these distinct phase transitions share a common, underlying mechanism.

The following picture emerges from our analysis: as long range correlations between gradients decay, residuals only interact with their immediate neighbours (onset of diffusion), leading to increased gradient variance (loss rate collapse) and translational symmetry breaking. In parallel, the growth of the principal eigenvalue or the NTK leads the principal eigenvector to memorize the distribution of influential points, as measured by accumulating gradients. In images, one influential class of points are edges, leading to their prominence in the principal eigenvector (NTK alignment).

In this study, we focused on SIREN models trained on a 2D super-resolution task using full-batch gradient descent. However, SIRENs are used in a variety of inverse problems, and it remains to be seen whether our observations extend to these settings. Future work may also explore the impact of different optimizers, such as ADAM [44], which adaptively adjusts learning rates and may influence the stability and divergence of the principal eigenvalue - a key factor in our study of NTK alignment.

This work has demonstrated that the NTK provides a rich theoretical tool for deriving and relating order parameters to understand training dynamics. We provide new methodology to rigorously study the influence of inductive biases, such as model architectures and hyper-parameters, on the underlying learning process and may have practical utility in diagnosing causes of poor learning outcomes.

# Acknowledgements

Supported by the University of Sussex Be.AI doctoral scholarship, funded by the Leverhulme Trust.

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

## Supplementary Materials

## A  Deferred Proofs

### A.1  Decomposition of the NTK over layers

Consider a feedforward neural network, denoted by $f(x) = h^{(L)} \circ \ldots h^{(1)}(x)$. We furthermore define:

$$z^{(l)} = [W^{(l)}]^\top h^{(l-1)} + b^{(l)} \tag{24}$$

$$h^{(l)} = \sigma(z^{(l)}) \tag{25}$$

In this way, we may calculate the parametric gradients as follows:

$$\nabla_{W^{(l)}} f = \frac{\partial f}{\partial z^{(l)}} [h^{(l-1)}]^\top \tag{26}$$

$$\text{vec}(\nabla_{W^{(l)}} f) = \left( I \otimes \frac{\partial f}{\partial z^{(l)}} \right) h^{(l-1)} \tag{27}$$

$$\text{vec}(\nabla_{W^{(l)}} f(x_i))^\top \text{vec}(\nabla_{W^{(l)}} f(x_j)) = h^{(l-1)}(x_i)^\top \left( I \otimes \frac{\partial f(x_i)}{\partial z^{(l)}}^\top \right) \left( I \otimes \frac{\partial f(x_j)}{\partial z^{(l)}} \right) h^{(l-1)}(x_j) \tag{28}$$

$$= h^{(l-1)}(x_i)^\top \left( I \otimes \frac{\partial f(x_i)}{\partial z^{(l)}}^\top \frac{\partial f(x_j)}{\partial z^{(l)}} \right) h^{(l-1)}(x_j) \tag{29}$$

$$= \left( \frac{\partial f(x_i)}{\partial z^{(l)}}^\top \frac{\partial f(x_j)}{\partial z^{(l)}} \right) \left( h^{(l-1)}(x_i)^\top h^{(l-1)}(x_j) \right) \tag{30}$$

The first term in this product defines functional similarity between points, while the second defines representational similarity. Thinking of each term as a separate kernel, the overall layer kernel - ie the product is defined via an AND operation. A similar formula holds for the other layers. For the biases, we have, more simply:

$$\nabla_{b^{(l)}} f = \frac{\partial f}{\partial z^{(l)}} \tag{31}$$

$$\nabla_{b^{(l)}} f(x_i)^\top \nabla_{b^{(l)}} f(x_j) = \frac{\partial f(x_i)}{\partial z^{(l)}}^\top \frac{\partial f(x_j)}{\partial z^{(l)}} \tag{32}$$

The full NTK is then given simply by:

$$K(x_i, x_j; \theta) = \sum_{l=1}^{N_l} \left( \frac{\partial f(x_i)}{\partial z^{(l)}}^\top \frac{\partial f(x_j)}{\partial z^{(l)}} \right) \left( 1 + h^{(l-1)}(x_i)^\top h^{(l-1)}(x_j) \right) \tag{33}$$

$$\equiv \sum_{l=1}^{N_l} K^{(l)}(x_i, x_j) \tag{34}$$

In particular, we have:

$$K(x, x; \theta) = \sum_{l=1}^{N_l} \left\| \frac{\partial f(x)}{\partial z^{(l)}} \right\|_2^2 \left( 1 + \|h^{(l-1)}(x)\|_2^2 \right) \tag{35}$$

Following the same logic, the full NTK is defined as an OR over all the layers. For INRs, these layers tend to be frequency separated, so that lower layers correspond to lower frequencies.

## A.2 Proof of Theorem 3.1: Diffusive Evolution of the Residuals

The motivation for our ansatz in equation 10 is the empirical form of the correlation function in equation 7. Written fully, we have:

$$K(x, x + u) \approx \|\nabla_\theta f(x)\|^2 \exp(-\|u\|^2/\xi^2(x)) + \|\nabla_\theta f(x)\|^2 c_\infty(x) \tag{36}$$

Thus the residuals evolve according to:

$$\dot{r}(x) = -\int du \, r(x+u) K(x, x+u) \tag{37}$$

$$\approx -\|\nabla_\theta f(x)\|^2 \left[ \int du \, \exp(-\|u\|^2/\xi^2(x)) r(x+u) - c_\infty(x) \int du \, r(x+u) \right] \tag{38}$$

$$= \|\nabla_\theta f(x)\|^2 \left[ \int du \, \exp(-\|u\|^2/\xi^2(x)) r(x+u) - \mu_r c_\infty(x) \text{Vol}(\mathcal{D}) \right] \tag{39}$$

When $\mu_r \equiv \mathbb{E}[r]$ and $c_\infty$ decay to zero, the second, background term in the above equation becomes dominated by local interactions. Thus, in Section 4, we will track the following order parameter:

$$\mu_r K_\infty \equiv |\mathbb{E}_x[r] \mathbb{E}_x[\|\nabla_\theta f(x)\|^2 c_\infty(x)]| \tag{40}$$

The order parameter is large in the Drift phase, and small in the Diffusion phase. In Section B, we overview the specifics of how we detect changes in the phase. For the remainder of this section, we analytically study kernels of the form:

$$K(x, x+u) = A(x)e^{-u^2/2\xi^2(x)} \tag{41}$$

$$= 2\pi\xi^2(x)A(x)\mathcal{N}(u; 0, \xi^2(x)I) \tag{42}$$

That is, kernels without a background term. Here, $\mathcal{N}(u; \mu, \Sigma)$ denotes the $d$-dimensional; multivariate Gaussian Distribution:

$$\mathcal{N}(u; \mu(x), \Sigma(x)) = \frac{1}{\sqrt{(2\pi)^d \det \Sigma(x)}} \exp\left(-\frac{1}{2}(u - \mu(x))^\top \Sigma^{-1}(x)(u - \mu(x))\right) \tag{43}$$

For our case, $d = 2$, and $\Sigma(x) = \xi^2(x)I$. The determinant of the covariance is as follows:

$$\det \Sigma(x) = (\xi^2)^2 \det I = \xi^4(x) \tag{44}$$

We now consider the integral of the following quadratic form:

$$\int du \, (u^\top H u) \, e^{-u^2/2\xi^2(x)} = 2\pi\xi^2(x) \int du \, (u^\top H u) \mathcal{N}(u; 0, \xi^2(x)I) \tag{45}$$

$$= 2\pi\xi^2(x)\mathbb{E}_{\mathcal{N}(u;0,\xi^2 I)}[u^\top H u] \tag{46}$$

$$= 2\pi\xi^2(x)\text{tr}(H\Sigma(x)) \tag{47}$$

$$= 2\pi\xi^4(x)\text{tr}(H) \tag{48}$$

Now, let's look at the following Taylor expansion:

$$r(x+u) \approx r(x) + u^\top \nabla_x r + \frac{1}{2}u^\top(\nabla_x^2 r)u \tag{49}$$

When integrating the above in equation 3, the second term vanishes, because it involves a product of symmetric and anti-symmetric functions. Thus, we have

$$\int du \, r(x+u)K(x, x+u) = A(x) \int du \, r(x+u)e^{-u^2/2\xi^2(x)} \tag{50}$$

$$= A(x) \int du \left[r(x)e^{-u^2/2\xi^2(x)} + \frac{1}{2}u^\top(\nabla_x^2 r)u \, e^{-u^2/2\xi^2(x)}\right] \tag{51}$$

Leveraging our result for the quadratic term, we have, finally:

$$\int du \, r(x+u)K(x, x+u) = 2\pi\xi^2(x)A(x)r(x) + \pi\xi^4(x)A(x)\text{tr}(\nabla_x^2 r) \tag{52}$$

$$= 2\pi\xi^2(x)A(x)r(x) + \pi\xi^4(x)A(x)\Delta^2 r \tag{53}$$

Thus, the diffusion equation becomes:

$$\dot{r} = -2\pi\xi^2(x)A(x)r(x) - \pi\xi^4(x)A(x)\Delta^2 r$$

### A.3 Proof of Theorem 3.2: Local Cauchy Approximation of the $C_{NTK}$

#### A.3.1 Notation and Derivation

We consider an arbitrary vector valued function $f(x)$, and consider the cosine of the angle between $f(x)$ and $f(x+u)$ for small displacements $u$. To ease notation, let us make use of the following shorthands:

$$a = f(x) \tag{54}$$
$$b = f(x+u) \tag{55}$$
$$c = b - a \tag{56}$$
$$J = \nabla_x a \tag{57}$$
$$D = \nabla_x ||a||^2 \tag{58}$$

To first order in $u$, we have:

$$b \approx a + u^\top J \tag{59}$$

$$c \approx u^\top J \tag{60}$$

$$||b||^2 \approx ||a + u^\top J||^2 \tag{61}$$

$$= ||a||^2 + u^\top D + ||u^\top J||^2 \tag{62}$$

Our goal is to discern the local behaviour of the cosine of the angle $\theta$ between $a$ and $b$ (as illustrated in Figure 5). To that end, our starting point is the law of cosines:

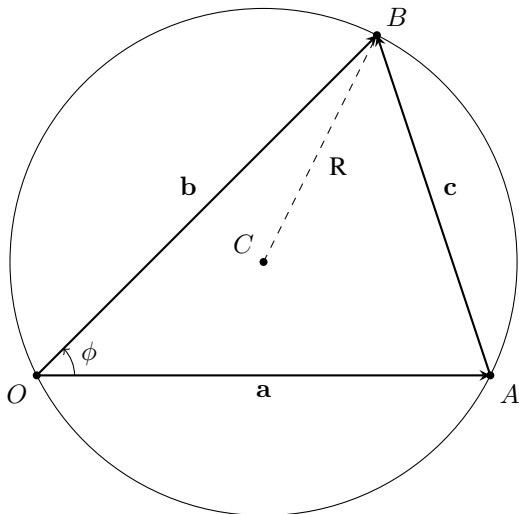

Figure 5: Triangle with vectors $\mathbf{a}$, $\mathbf{b}$, and $\mathbf{b} - \mathbf{a}$, inscribed in a circumcircle.

$$\cos \phi = \frac{||a||^2 + ||b||^2 - ||c||^2}{2||a|| \, ||b||} \tag{63}$$

$$\approx \frac{2||a||^2 + u^\top D}{2||a||^2} \left( 1 + \frac{u^\top D}{||a||^2} + \frac{||u^\top J||^2}{||a||^2} \right)^{-\frac{1}{2}} \tag{64}$$

To proceed, note that, for small scalar $\epsilon$, we have the following identity:

$$(1 + \epsilon)^{\frac{1}{2}} \approx 1 + \frac{\epsilon}{2} - \frac{\epsilon^2}{8} \tag{65}$$

Thus:

$$\cos \phi \approx \frac{2||a||^2 + u^\top D}{2||a||^2 + u^\top D + ||u^\top J||^2 - \frac{1}{16||a||^2}(u^\top D)^2} \tag{66}$$

$$\tag{67}$$

For the NTK, where we will have $a = \nabla_\theta f$, $||a||$ is so large that we may ignore the term of order $||a||^{-2}$. We illustrate our approximation in Figure 6.

### A.3.2 Specialization for Feed Forward Neural Networks

We want to consider the case where, per our previous derivation, $a = \nabla_\theta f$. This procedure is straightforward for the biases. For the weights $W_{ij}^{(l)}$, we have:

$$\frac{\partial f(x; \theta)}{\partial W_{ij}^{(l)}} = \frac{\partial f(x; \theta)}{\partial z_i^{(l)}} h_j^{(l-1)}(x; \theta) \tag{68}$$

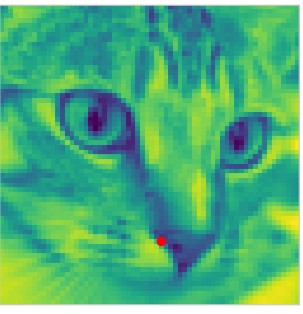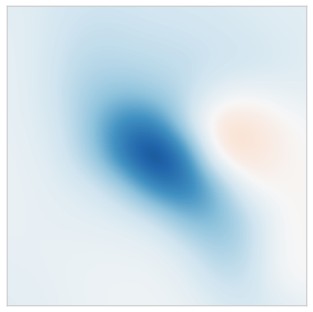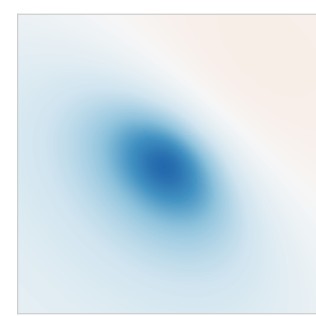

Figure 6: **Cauchy Approximation of the Cosine NTK**. Left: Sample image, and test point $x = A$. Middle: visualization of $C_{NTK}(x, x+u)$ in the vicinity of the point $A$ for small separations $u$. Right: the Cauchy approximation, capturing both the range, orientation, and the local minima of the true $C_{NTK}$.

Therefore:

$$\frac{\partial f^2(x;\theta)}{\partial x_m \partial W_{ij}^{(l)}} = \frac{\partial^2 f(x;\theta)}{\partial x_m \partial z_i^{(l)}} h_j^{(l-1)}(x;\theta) + \frac{\partial f(x;\theta)}{\partial z_i^{(l)}} \frac{\partial h_j^{(l-1)}(x;\theta)}{\partial x_m} \tag{69}$$

$$\triangleq (J_z^{(l)})_{im} h_j^{(l-1)} + (J_h^{(l-1)})_{jm} \partial_{z_i^{(l)}} f \tag{70}$$

Before proceeding, let us note that the following holds:

$$\sum_i (J_z^{(l)})_{im} (\partial_{z_i^{(l)}} f) = \frac{1}{2} \partial_{x_m} ||\nabla_{z^{(l)}} f||^2 \tag{71}$$

$$\sum_i (J_h^{(l)})_{im} h_i^{(l)} = \frac{1}{2} \partial_{x_m} ||h^{(l)}||^2 \tag{72}$$

The covariance matrix in our Gaussian approximation is thus given by:

$$H_W^{(l)} = \sum_{i,j} \frac{\partial f^2}{\partial x_m \partial W_{ij}^{(l)}} \frac{\partial f^2}{\partial x_n \partial W_{ij}^{(l)}} \tag{73}$$

$$= \sum_{i,j} (h_j^{(l-1)})^2 (J_z^{(l)})_{im} (J_z^{(l)})_{in} + (\partial_{z_i^{(l)}} f)^2 (J_h^{(l-1)})_{jm} (J_h^{(l-1)})_{jn} \tag{74}$$

$$+ (J_z^{(l)})_{im} (\partial_{z_i^{(l)}} f)(J_h^{(l-1)})_{jn} h_j^{(l-1)} + (J_z^{(l)})_{in} (\partial_{z_i^{(l)}} f)(J_h^{(l-1)})_{jm} h_j^{(l-1)}$$

$$= ||h^{(l-1)}||^2 J_z^{(l)} J_z^{(l)\top} + ||\nabla_{z^{(l)}} f||^2 J_h^{(l-1)} (J_h^{(l-1)})^\top \tag{75}$$

$$+ \frac{1}{4} \nabla_x ||h^{(l-1)}||^2 \otimes \nabla_x ||\nabla_{z^{(l)}} f||^2 + \frac{1}{4} \nabla_x ||\nabla_{z^{(l)}} f||^2 \otimes \nabla_x ||h^{(l-1)}||^2$$

The contribution from the bias is comparatively simple:

$$H_b^{(l)} = J_z J_z^\top \tag{76}$$

## A.4 Proof of Corollary 3.2.1: Obtaining the Correlation Lengthscale from the Cauchy Approximation

To determine the level sets of the Cauchy Approximation, we must solve:

$$C_{NTK}(x, x+u) = \frac{2a_x^2 + u^\top D_x}{2a_x^2 + u_x^\top D + u^\top H_x u} = c \tag{77}$$

Rearranging, and collecting terms, we have:

$$2a_x^2 + u^\top D_x - c(2a_x^2 + u_x^\top D + u^\top H_x u) = 0 \tag{78}$$

$$\Rightarrow 2(1-c)a_x^2 - c\left(-\frac{1-c}{c}u^\top D_x + u^\top H_x u\right) = 0 \tag{79}$$

$$\Rightarrow u^\top H_x u - \frac{1-c}{c}u^\top D_x = \frac{2(1-c)}{c}a_x^2 \tag{80}$$

$$\Rightarrow \left(u - \frac{1-c}{2c}H^{-1}D\right)^\top H\left(u - \frac{1-c}{2c}H^{-1}D\right) - \frac{(1-c)^2}{4c^2}D^\top H^{-1}D = \frac{2(1-c)}{c}a_x^2 \tag{81}$$

$$\Rightarrow \frac{\left(u - \frac{1-c}{2c}H^{-1}D\right)^\top H\left(u - \frac{1-c}{2c}H^{-1}D\right)}{\frac{2(1-c)}{c}a_x^2 + \frac{(1-c)^2}{4c^2}D^\top H^{-1}D} = 1 \tag{82}$$

This is the equation of an ellipse centred at $u = \frac{1-c}{2c}H^{-1}D$, and with shape matrix:

$$\Sigma_{shape} = \frac{H}{\frac{2(1-c)}{c}a_x^2 + \frac{(1-c)^2}{4c^2}D^\top H^{-1}D} \tag{83}$$

The area of this ellipse is (noting that $H$ is a 2x2 matrix):

$$A_{ellipse} = \frac{\pi}{\sqrt{\det \Sigma_{shape}}} \tag{84}$$

$$= \frac{1}{\sqrt{\det H}}\left(\frac{2(1-c)}{c}a_x^2 + \frac{(1-c)^2}{4c^2}D^\top H^{-1}D\right) \tag{85}$$

The correlation lengthscale is then obtained from:

$$\xi = \sqrt{A_{ellipse}/\pi} \tag{86}$$

## A.5  Proof of Corollary 3.2.3: Minimum Value of $C_{NTK}$

We consider minimizing the following function:

$$f(u) = \frac{Q(u)}{P(u)} \tag{87}$$

$$Q(u) = 2a^2 + u^\top D \tag{88}$$

$$P(u) = Q(u) + u^\top H u \tag{89}$$

Here, $H$ is non-degenerate and positive definite. Thus:

$$\frac{\partial f}{\partial u} = \frac{\partial_u Q P - Q \partial_u P}{P^2} = 0 \tag{90}$$

$$\implies \partial_u Q P = Q \partial_u P \tag{91}$$

Thus:

$$(u^\top H u)D = (4a^2 + 2u^\top D)Hu \tag{92}$$

Clearly $u = 0$ is a solution, and knowing that our expression locally approximates the cosine, we expect this to be a maximum. To find the other solution, which will be a minima, we take the dot product of both sides of the above equaiton with $u$. After simplifying, we obtain:

$$u^\top D = -4a^2 \tag{93}$$

If we insert this into equation 92, we get:

$$(u^\top H u)D = -4a^2 Hu \tag{94}$$

$$\Rightarrow (u^\top H u)H^{-1}D = -4a^2 u \tag{95}$$

$$\Rightarrow (u^\top H u)(D^\top H^{-1}D) = 16a^4 \tag{96}$$

$$\Rightarrow u^\top H u = \frac{16a^4}{DH^{-1}D} \tag{97}$$

Armed with an expression for $u^\top D$ and $u^\top Hu$, we derive the following formula for the min:

$$f_{min} = \frac{2a^2 + u^\top D}{2a^2 + u^\top D + u^\top Hu}\bigg|_{u=\text{argmin}f} \tag{98}$$

$$= \frac{2a^2 - 4a^2}{2a^2 - 4a^2 + \frac{16a^4}{DH^{-1}D}} \tag{99}$$

$$= \frac{DH^{-1}D}{DH^{-1}D - 8a^2} \tag{100}$$

## A.6   Relating Loss Gradient Variance to the NTK

Our goal is to quantify the amount of noise in the gradients of the local loss $\mathcal{L}(x_i) = \frac{1}{2}r(x_i;\theta)^2$. We have, in terms of the Jacobian $J_{ip} = \partial_{\theta_p} f(x_i)$, the following sample matrix for the gradients:

$$G = RJ \tag{101}$$

Here we have defined:

$$R = \text{diag}(r) \tag{102}$$

For a dataset with $N$ samples, the sample mean and covariance are given by:

$$\mu = \frac{1}{N}G^\top 1_N \tag{103}$$

$$= \frac{1}{N}J^\top r \tag{104}$$

$$C = \frac{1}{N}J^\top R^2 J - \mu\mu^\top \tag{105}$$

From the cycle property of the trace, we have:

$$\text{tr}(J^\top R^2 J) = \text{tr}(R^2 JJ^\top) \tag{106}$$

$$= \text{tr}(R^2 K_{NTK}). \tag{107}$$

We also have:

$$\text{Tr}(\mu\mu^\top) = ||\mu||^2 \tag{108}$$

$$= \frac{1}{N^2}r^\top JJ^\top r \tag{109}$$

$$= \frac{1}{N^2}r^\top K_{NTK}r \tag{110}$$

Thus the variance of the loss gradients is given by:

$$\sigma_\theta^2 = \frac{1}{N}\text{Tr}(R^2 K_{NTK}) - \frac{1}{N^2}r^\top K_{NTK}r \tag{111}$$

# B   Experimental Details

## B.1   Model Training

All our SIREN models are trained on the images shown in Figure 7, which we obtain through the python package scikit-image [45], and the ImageNet dataset [46]. These images are down-sampled to a resolution of $64 \times 64$ for training, but as a validation task, we track the reconstruction error on the images downsampled to $256 \times 256$ resolution. Our SIREN models are implemented using Pytorch [47], and trained using NVIDIA RTX A6000 48GB GPUs for 10000 epochs, using full batch gradient descent with a learning rate of 1e-3. In our experimental sweeps, we consider the following ranges:

- Random seeds from interval $[0, 5]$.
- Width from set $\{64, 128\}$.
- Depth from set $\{3, 4, 5\}$.
- $\omega_0$ from set $\{15, 30, 60, 90\}$.

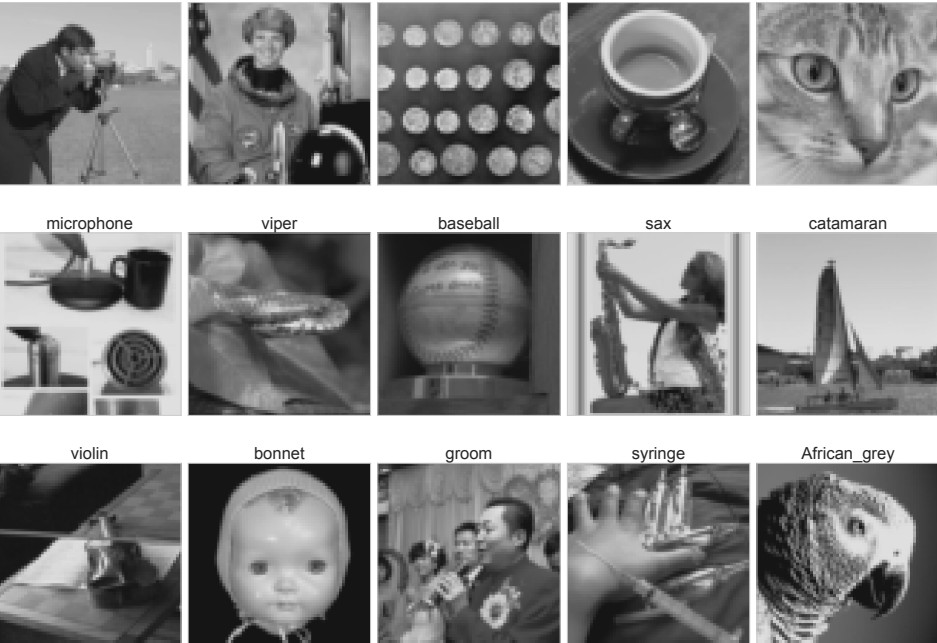

Figure 7: **Fifteen of the thirty images used for training INRs**

## B.2 Order Parameter Estimation

**Analytical Order Parameters**: To compute the NTK, we use a manual implementation of back-propagation to compute the gradients $\nabla_{z^{(l)}} f(x)$ for each layer, along with the hidden activations $h_{(l)}(x)$. The NTK is then constructed efficiently using the decomposition across layers outlined in Section A.1. To evaluate the local structure components $a$ and $D$ defined in Theorem 3.2, we obtain the spatial gradients using functorch [48]. We also assemble the $H$ defined in Theorem 3.2 in this way, except we leverage the decomposition outlined in Section A.3.2 to streamline this process, and occupy less memory.

**Empirical Order Parameters**: Below we describe the estimation procedure for each of the empirical order parameters.

- To estimate the correlation functions empirically, we group pairs of datapoints into 50 bins based on a uniform division of the range of distances. Based on the coordinate range, the minimum distance is 0, and the maximum distance is $2\sqrt{2}$. Within each bin, we evaluate the mean of the $C_{NTK}$, defining $c(\epsilon)$. Based on these groups, we estimate our order parameters as follows:

  - To estimate the asymptotic value $c_\infty$, we compute the mean value of $c(\epsilon)$ over the last ten bins (corresponding to points with the furthest separation).
  - Given the asymptotic value, we rescale all $c(\epsilon) \to \tilde{c}(\epsilon) = \frac{c(\epsilon)}{1 - c_\infty}$, and then use linear interpolation to find the value of $\epsilon$ for which $\tilde{c}(\epsilon) = 0.5$, the FWHM. We then have $\xi_{corr} = \frac{\text{FWHM}}{\sqrt{2 \ln 2}}$.

- As an additional measure of the correlation length-scale (which we will use in Appendix D), we may calculate the number of points $N_C$ for which $C_{NTK}$ is greater than some cutoff (we use $\frac{1}{2}(1 + c_\infty)$). The effective correlation lenght-scale is then given by $\sqrt{N_C dA / \pi}$, where $d\tilde{A}$ is the area of the coordinate grid cells. We denote this estimate $\xi_{FWHM}$.

- To estimate $\text{AUC}(|v_0|, \nabla I)$, the ground truth edges are identified using the Canny Edge Detector distributed through scikit-image [45]. We then evaluate the Area Under the Receiver Operating Characteristic Curve (ROC AUC) using the implementation in scikit-learn [49]. The principal eigenvector $v_0$, and the principal eigenvalue $\lambda_0$, are both computed using our

own implementation of the Randomized Singular Value Decomposition built with pytorch [47], using 3 iterations and 10 oversamples.

- To evaluate the Centred Kernel Alignment, in order to prevent zero modes from obscuring alignment, the following centred-variant of the normalized Hilbert-Schmidt Information Criterion (HSIC) is employed:

$$\text{CKA}(K, K') = \frac{\text{Tr}(K_c K_c')}{\sqrt{\text{Tr}(K_c K_c)\text{Tr}(K_c' K_c')}} \tag{112}$$

Here, $K_c$ denotes that a centrering operation has been applied, and is defined as:

$$K_c = (I - \frac{1}{n}11^\top)K(I - \frac{1}{n}11^\top) \tag{113}$$

For both $K_X$ and $K_Y$, we use bandwidths $\kappa = 0.1$.

- To determine the residual correlations in Table 1, we randomly sample (and flatten) 15000 $15 \times 15$ patches from the validation residuals, and compute the pearson correlation matrix. We then record the mean correlation between all pixels in the patch and the patch centre.

**Identifying Critical Points**:

- For the gradient variance $\sigma_\theta^2$, the loss rate $\dot{L}_{\text{eval}}$, and the background contribution $\mu_r K_\infty$ the location, and confidence region, for the critical points are identified using the peak detection algorithm distributed through scipy.signal [50]. For the gradient variance, we filter for peaks with a prominence of $0.2$, loss rate we use $0.4$, and for the background we use $0.2$. In the case where multiple peaks are found, we select the peaks which appear closest in time. Finally, for $\mu_r K_\infty$, the phase transition occurs not at the peak itself, but after the signal decays to zero. Thus we use as confidence region the interval between the identified peak and the right-most boundary.

- For the $\min C_{NTK}$, we linearly interpolate to find the time $t$ where $\min C_{NTK}$ crosses $0$. To compute the confidence interval, we also track the cumulative std of $\min C_{NTK}$, denoted $\epsilon(t)$. We then use the same linear interpolation strategy to find the times where $\min C_{NTK} = \epsilon(t)$ and $\min C_{NTK} = -\epsilon(t)$.

- For all other parameters, we fit a sigmoid using the curve fitting function from scipy.optimize, with the default settings. The curve we fit has the form:

$$f(x; A, B, \mu, w) = A + (B - A)\left(1 + e^{-(x-\mu)/w}\right)^{-1} \tag{114}$$

We identify the time $t = \mu$ with the critical point, with confidence region defined by $\mu \pm 2w$. For MAG-MA, where the goal is to detect deviation from zero, we fit this sigmoid to the cumulative STD.

## C Occurrence Rates of Phase Transitions

### C.1 Impact of Image Features

There are three main cases in which a critical point cannot be reliably identified in an order parameter trajectory:

1. Peaks in the gradient variance $\sigma_\theta$ may be absent, or not prominent enough, to be detected using a standard peak detector.

2. A zero-crossing cannot be found for the $\min C_{NTK}$ because, at initialization, it is already less than $0$.

3. The order parameters do not saturate, and thus, are poorly represented as sigmoids. This is really only a problem for the edge alignment $\text{AUC}(|v_0|, \nabla I)$ and the task alignment $\text{CKA}(K_Y, K_{NTK})$. In the latter case, in some trials we see CKA steadily decrease after the inflection point of the loss. Numerically, we omit runs where the mean squared error of the fitted sigmoid is greater than $0.01$.

Table 2: **Proportion of runs with errors**: Frequency at which runs were ommitted in constructing Figure 3, as a function of depth and bandwidth $\omega_9$.

| depth | $\omega_0$ | AUC($|v_0|, \nabla I$) | CKA($K_Y, K_{NTK}$) | $\sigma_\theta$ | min $C_{NTK}$ |
|---|---|---|---|---|---|
| 3 | 90 | 0.570 | 0.293 | 0.007 | 1.000 |
| 3 | 60 | 0.447 | 0.177 | 0.023 | 1.000 |
| 3 | 30 | 0.503 | 0.157 | 0.093 | 0.700 |
| 3 | 15 | 0.643 | 0.773 | 0.583 | 0.600 |
| 4 | 90 | 0.117 | 0.067 | 0.000 | 0.500 |
| 4 | 60 | 0.067 | 0.013 | 0.000 | 0.500 |
| 4 | 30 | 0.043 | 0.057 | 0.037 | 0.500 |
| 4 | 15 | 0.257 | 0.693 | 0.170 | 0.343 |
| 5 | 90 | 0.017 | 0.017 | 0.000 | 0.470 |
| 5 | 60 | 0.037 | 0.003 | 0.000 | 0.203 |
| 5 | 30 | 0.033 | 0.060 | 0.000 | 0.083 |
| 5 | 15 | 0.070 | 0.643 | 0.070 | 0.093 |

The occurence rates, as a function of the hyperparameters used, are shown in Table 2. It is important to note, phase transitions may still occur even during these failure modes - the shift in the order parameter may be simply too weak[3] to be identified by the change detection algorithm outlined in Section B.2. It is for this reason that we employ multiple order parameters to identify the same transition (ex the min $C_{NTK}$ and the gradien variance $_\theta$). Nevertheless, it is instructive to identify what properties of image datasets may be used to predict the aforementioned failure modes. To this end, for each experimental run, we determine if any of the previously mentioned failure modes has occurred, and then record the frequency of success for each image studied. In Figure 8, we see that these frequencies correlate with the complexity of the image, as measured the variance of the spatial gradient magnitudes $||\nabla_x I||$. Namely, we see that more complex images result in sharper peaks of the parameter gradient variance $\sigma_\theta$, but collapse of the kernel alignment as measured by CKA($K_Y, K_{NTK}$). This is reflected in their strong negative/positve spearman correlations. These same properties correlate strongly with the best model performance achieved across all hyperparams (left of Figure 8). These correlations give additional support to the mechanism described in Section 3.3, whereby SIREN models struggle to fit edges as they have sharp gradients. Finally, we note that the image complexity seems to have little impact on the error rates for the edge alignment AUC($|v_0|, \nabla I$) (spearman correlation -0.017) and the minimum value of the $C_{NTK}$ (spearman correlation -0.1320). By contrast, these parameters are more sensitive to the model architecture.

## C.2    Additional Figures: Impact of Hyperparameters

The broad effects of varying depth and $\omega_0$ on AUC($|v_0|, \nabla I$) are summarized in Table 1. To gain deeper insight into how these parameters influence the principal eigenvector, we examine the two case studies illustrated in Figures 10-11. A lower $\omega_0$, by broadening the correlation lengthscale, inducing a smoothing effect, retaining only the sharpest edges. Increasing depth also removes noise.

---

[3]Also note, even when AUC($|v_0|, \nabla I$) is weak, edges are still visible in the principal eigenvector, as seen in Figure 9

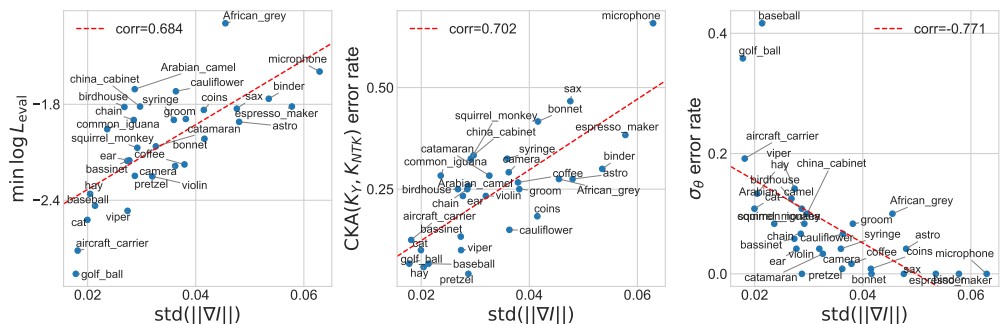

Figure 8: **Image Complexity Affects Detection of Phase Transitions**. We measure the image complexity according the standard deviation of the magnitude of the spatial gradients ($||\nabla_x I||$). Dashed red line indicates line of best fit. Legend records spearman correlation. Left: higher complexity images are positively correlated with higher losses (and therefore, worse performance). Middle: higher complexity images do not saturate the target kernel alignment, causing errors in our sigmoidal fits. Right: higher complexity images lead to sharper peaks in the paramter gradient variance, making their identification easier.

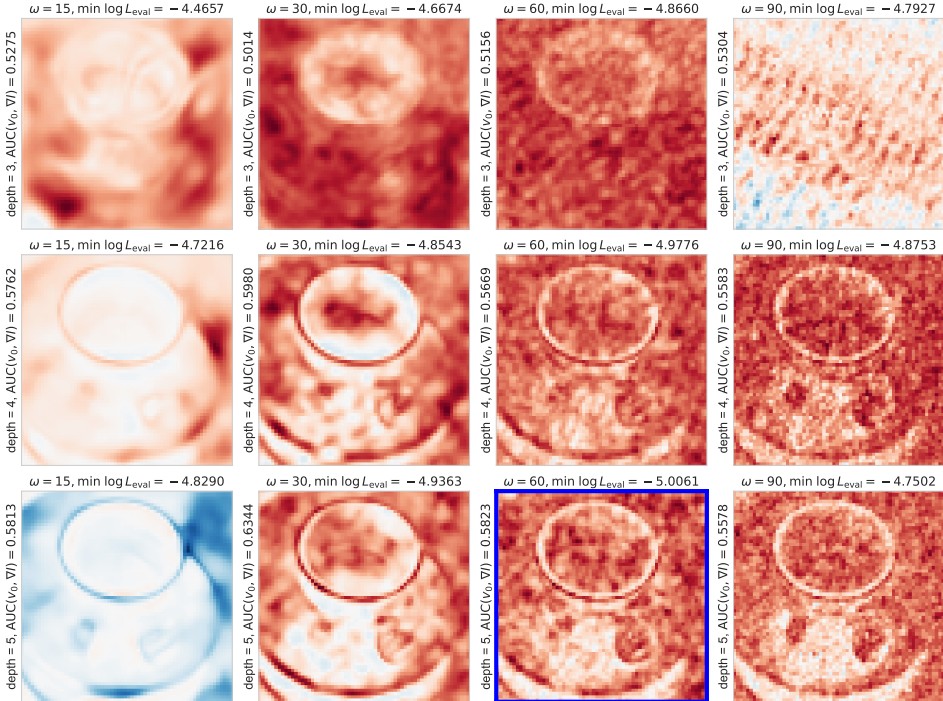

Figure 9: **Variation in NTK Alinment with Hyperparameters (coffee)**. Principle eigenvectors of the NTK at the end of training. Best performing architecture highlighted in blue.

# D   Comparison with ReLU Activations

To justify our focus on sinusoidal neural networks, in this section we examine the learning dynamics of ReLU-MLPs, based on the positional encoding scheme used in [51]. The positional encoding layer is kept static, and we pre-compute the nyquist frequencies corresponding to our image size ($64 \times 64$), as is done in [23]. We denote this architecture ReLU-PE. All other architectural choices are identical to those described in Appendix B. We observe a number of differences between SIRENs and ReLU-PEs (visualized in Figure 13):

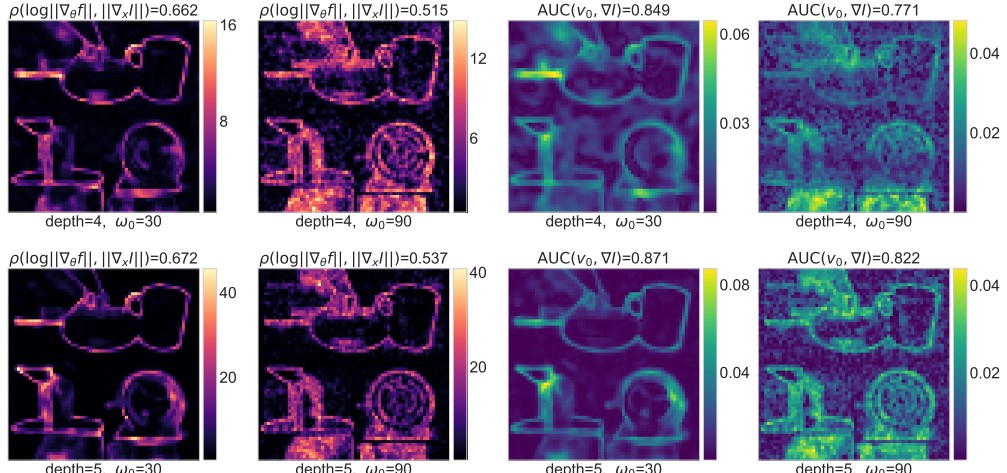

Figure 10: **Effect of Hyperparameters on Edge Alignment**: Reproduction of Figure 4 for the microphone dataset

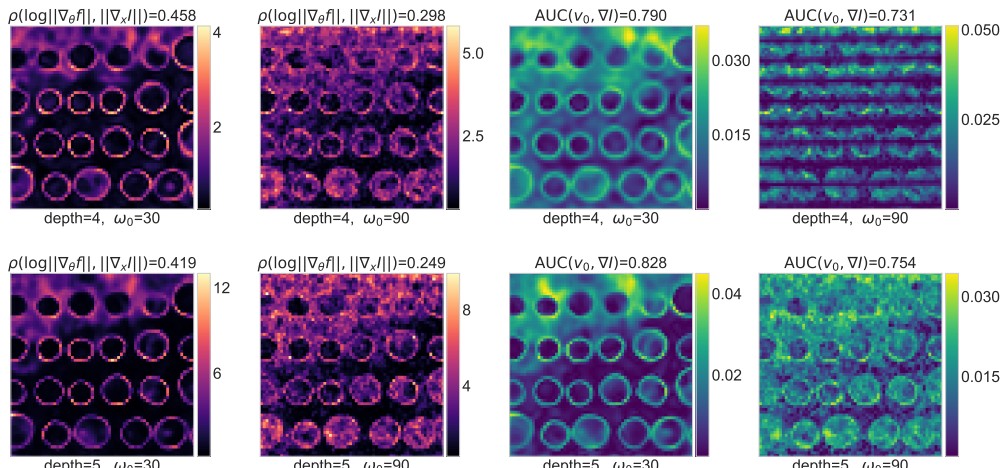

Figure 11: **Effect of Hyperparameters on Edge Alignment**: Reproduction of Figure 4 for the coins dataset

- Firstly, SIREN models exhibit strong locality: over the course of training, the asymptotic value of the $C_{NTK}$ decays to 0, whereas it grows in ReLU-PE models. What's more, the range of interaction as measured by $\xi_{FWHM}$ is larger in ReLU-PE models. An example comparing the correlation functions for both architectures is shown in Figure 12.

- Secondly, learning is much slower in ReLU-PE models than it is in SIRENs. One explanation for this is that there is more gradient confusion [11], that is, the minimum value of the $C_{NTK}$ is lower. In particular, $\min C_{NTK}$ is less than zero across all ReLU-PE runs, so that these models are always operating in the "slow" phase of learning.

- The principal eigenvalue $\lambda_0$ of the NTK grows to be orders of magnitude larger for SIREN models than for ReLU-PE models. That said, tangent kernel alignment still occurs in ReLU-PE models, it is just a much slower process. In Figure14, we train a 7-layer deep, 128-unit wide MLP full-batch with a learning rate of $1e-3$ for 250k epochs, varying only the activation function. To reach the edge-alignment achieved by a SIREN model after 453 epochs, the ReLU-PE model must train for 239986 epochs. We also see that more of the edges are present in the principal eigenvector of the SIREN model's NTK.

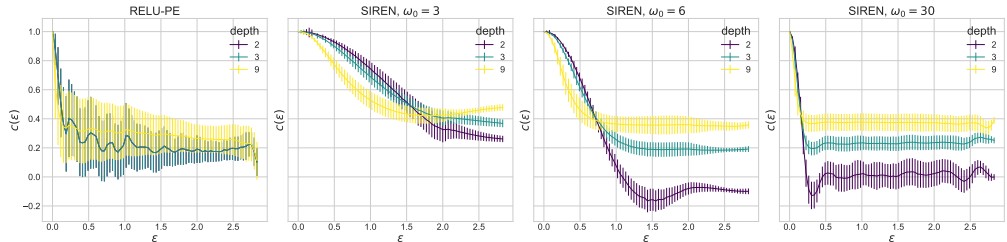

Figure 12: **Effect of Hyperparameters on Correlation Functions At Initialization**: In ReLU-PE models, the Gaussian approximation of the $C_{NTK}$ correlation function is poor for all depths, due to high-variance, long range interactions. By contrast, for SIREN models, there is much less variance, and the range of the interactions shrinks for increasing $\omega_0$.

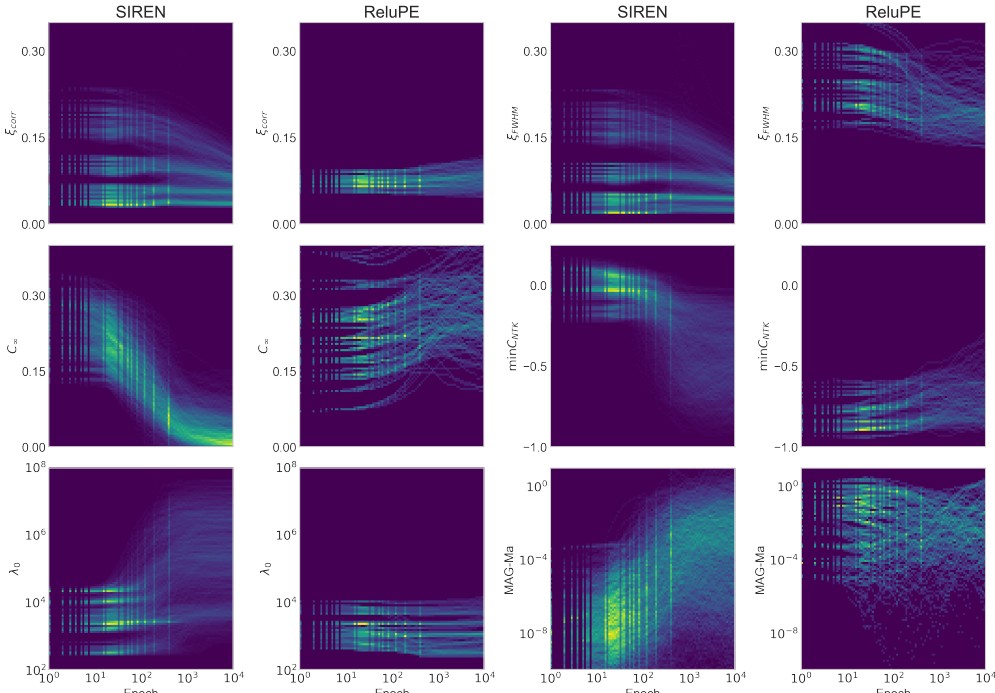

Figure 13: **Learning Trajectories for SIREN and ReLU-PE models**: Histograms visualizing the distribution of various order parameters throughout training. See Section B for full details on models and datasets used.

- At initialization, MAG-Ma is orders of magnitude lower for SIREN models than for Relu-PE models, indicating the latter are already operating in a phase where translational symmetry is broken.

In summary, while ReLU-PE models exhibit Neural Tangent Kernel alignment, it is a much slower, non-local process, that does not coincide with loss-rate collapse or translational symmetry breaking.

# E   Implications of Local Image Structure on Feature Learning

## E.1   On the Relationship Between Structure Tensors and Tangent Kernels

We are now positioned to elucidate the features learned during NTK alignment. As proposed in Section 3.3, the local structure of the NTK adapts to the spatial variations in parameter gradients. In this section, we delve into the spectral consequences of this adaptation. We contend that the principal eigenvectors evolve into edge detectors, resembling the auto-correlation structure tensors

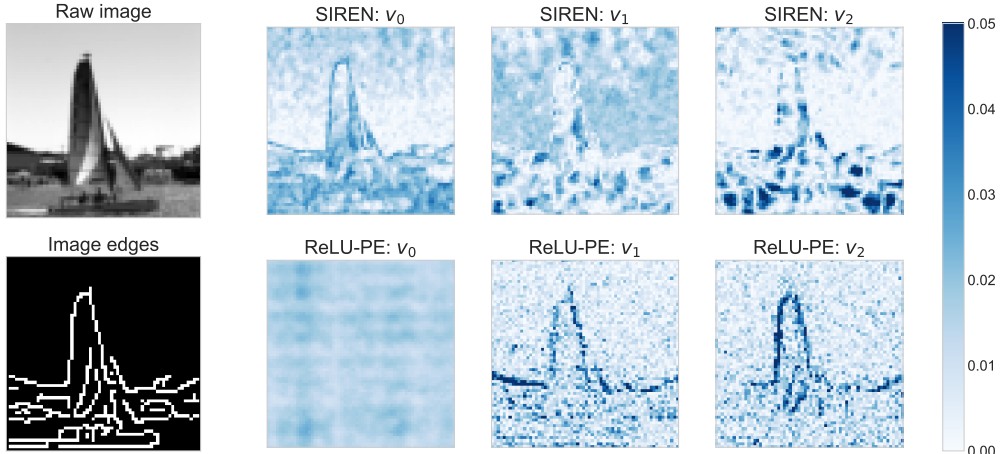

Figure 14: **NTK alignment in SIREN and ReLU-PE models**: The principal eigenvectors of the NTK at the end of training. Final $\text{AUC}(|v_1|, \nabla I)$ for the ReLU-PE is 0.754, whereas $\text{AUC}(|v_0|, \nabla I)$ for the SIREN model is 0.804. The training time required to achieve an edge-alignment score greater than 0.75 for the SIREN model was 453 epochs, whereas for the ReLU-PE model it was 239986 epochs.

commonly employed in traditional computer vision. This observation reinforces the concept of translation symmetry breaking: in computer vision, the utility of auto-correlation structure tensors stems from the premise that the most informative features are those that minimize redundancy. The auto-correlation function quantifies this through metrics of translational symmetry breaking.

Per the discussion in Section 3.3, the principal eigenvector is closely related to the auto-correlation function. By leveraging the decomposition of the NTK in equation 30, we may relate to the features considered in computer vision. Let us define:

$$w^{(l)}(u; x) = 1 + h^{(l-1)}(x)^\top h^{(l-1)}(x + u) \tag{115}$$

so that the largest contribution comes from the immediate neighbourhood of $x$. This motivates us to perform a Taylor expansion of the remaining terms as follows:

$$K_l 1 = \sum_u K_l(x, x + u) \tag{116}$$

$$= \sum_u w_l(u; x) \sum_d \frac{\partial f(x)}{\partial z_{ld}} \frac{\partial f(x + u)}{\partial z_{ld}} \tag{117}$$

$$\approx \sum_u w_l(u; x) \sum_d \left( \frac{\partial f(x)}{\partial z_{ld}} \frac{\partial f(x)}{\partial z_{ld}} + h.o.t \right) \tag{118}$$

$$= \text{tr}(A_l(x_i)) + h.o.t \tag{119}$$

Here, $A_l$ denotes the structure tensor used in the Harris-Corner detector [52]. Accordingly, we see that $K1$ - and thus, the principle eigenvector - assess the extent of local translational symmetry disruption near a point $x$. This principle underlies feature selection in computer vision, a concept mirrored in NTK feature learning, as evidenced by the principal eigenvectors that are predominantly maximized around dataset edges and corners.

It is crucial to highlight that $A_l$ pertains to the structure tensor of a specific layer $l$. Collectively, the entire DNN's NTK facilitates feature selection across a scale pyramid.

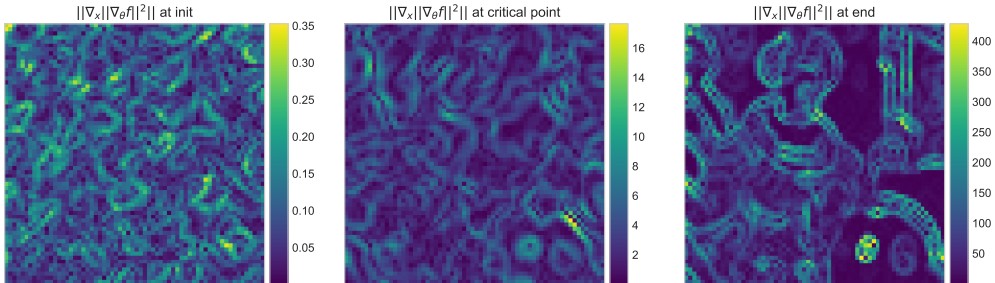

Figure 15: Evolution of spatial variation of the parameter gradients. At initialization, there is a very small amount of variance (note the scale of the variations). As the variance grows, translational symmetry is broken, and a dynamical phase transition occurs.

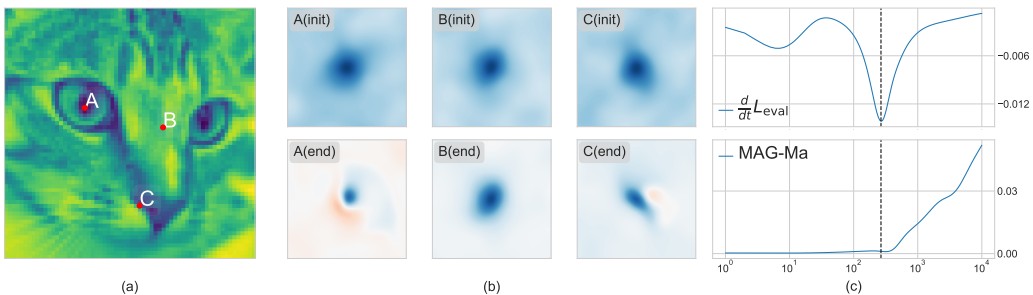

Figure 16: **Evolution of the Cosine NTK**: We visualize $C_{NTK}(x, x + u)$ around three points $x \in \{A, B, C\}$ for small separations $u$. At initialization, $C_{NTK}$ locally resembles an isotropic, translation-invariant RBF. However, as training progresses, these symmetries are broken. MAG-Ma (described in Section E.2) is an order-parameter that monitors the original symmetry, and changes at the critical point.

## E.2 MAG-MA: Order Parameters From Translational Symmetry Breaking

While previous sections have focused on bottom-up construction of order parameters, this section adopts a top-down approach rooted in symmetry principles. In Sections 3.1-3.4, we expressed several order parameters in terms of the parameters $a, D, H$, characterizing the local structure of the $C_{NTK}$. Notably, each of these parameters is now a function of the spatial variation of the parameter gradients, whose evolution is showcased in Figure 15. This suggests it is a translation symmetry which is broken at the phase transition. Indeed, from Figure 16, we observe the $C_{NTK}$ is an approximately stationary, isotropic kernel - a desirable property for INRs [30]. As such, the Kernel exhibits no bias for location or direction. Over the course of training, we may monitor the emergence of such a bias with the following metric :

$$||\mathbb{E}_x[\nabla_x \log ||\nabla_\theta f||^2]||^2 = ||\mathbb{E}_x[D_x/a_x^2]||^2 \tag{120}$$

We refer to this statistic as **MAG-Ma**: the **M**agnitude of the **A**verage **G**radient of the Log Gradient-Field **Ma**gnitudes. Intuitively, this order parameter captures the statistical preference for a spatial direction in the dataset. The evolution of this quantity is plotted in Figure 16, and its alignment with the other order parameters is shown in Figure 3. We see that throughout the Fast Phase of training (before the peak in the loss rate $\dot{L}_{eval}$), the local structure of the $C_{NTK}$ is statistically translation invariant, and MAG-Ma is close to zero. However, just after the critical point, it grows rapidly - coinciding with the edge memorization described in Section 3.3.

## F Evaluating Fidelity of Approximation

### F.1 Local Structure of the NTK

As described in Section 3.1, INRs are often carefully designed to ensure a diagonally dominant NTK [30, 31, 32]. In higher dimensions, diagonal dominance is equivalent to a bias towards local

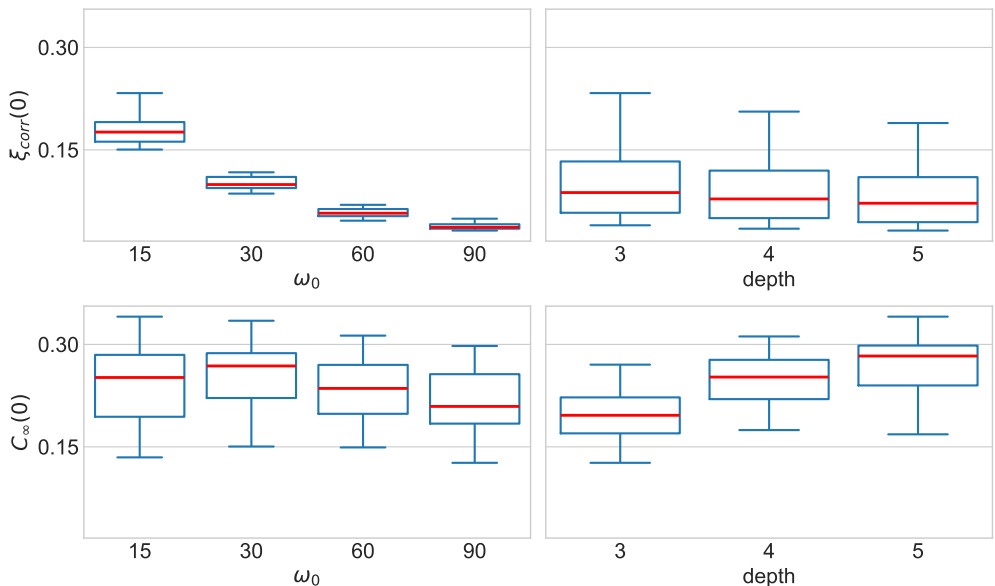

Figure 17: **Hyperparameters Affect Local NTK Structure**. Boxplots visualizing the distribution of structural parameters for the $C_{NTK}$. Top row: variation in the initial correlation lengthscale $\xi_{corr}(0)$. Bottom row: variation in the initial asymptotic value of the $C_{NTK}$ ($C_\infty(0)$).

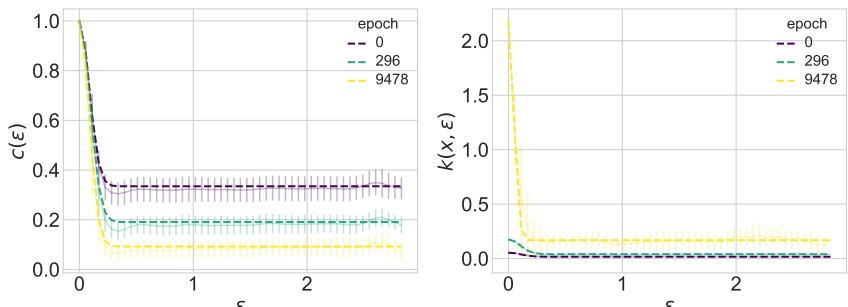

Figure 18: Visualization showing the empirical correlation function for the normalized parameter gradients. On the left-hand side is the global correlation-function for the $C_{NTK}$. On the right is the local-correlation function for the $K_{NTK}$ around a test point $x$. Dashed lines show fitted Gaussian approximation, and error bars show variance across dataset. Over the course of training, both the global correlation lengthscale $\xi_{corr}$, and the terminal value $c_\infty$, evolve.

interactions. We see in Figure 17 the hyperparameters that most affect this local structure: we observe that, while depth has a small impact on the initial correlation lengthscale $\xi_{corr}(0)$, higher values of $\omega_0$ cause the $C_{NTK}$ to become dramatically more localized. The converse is true for the asymptotic value $C_\infty(0)$: $\omega_0$ has a minor effect, but increasing depth leads to stronger interactions across large distances.

Beyond initialization, in Figure 18, we examine the evolution of the correlation function for a five-layer deep, 128-unit wide SIREN model on a $128 \times 128$ grayscale image of a cat, with bandwidth $\omega_0 = 30$. Emprirically, we see that the Gaussian approximation described in Section 3.1 remains valid across training, with the asymptotic value $c_\infty$ of the $C_{NTK}$ decaying to zero.

### F.2 Cauchy Approximation

To ascertain the fidelity of the Cauchy Approximation, we estimate the Pearson correlation between the true values of the correlation lengthscale $\xi$ and $\min C_{NTK}$, and the prediction based only on the local model. We choose this metric because the identification of critical points is insensitive to linear

Table 3: **Fidelity of Cauchy Approximation**: Pearson correlation between the true order parameter and predictions using the local Cauchy Approximation. Mean and standard deviation are calculated over the spread of models and datasets described in Section B.

| depth | $\omega_0$ | $\xi$ | $\min C_{NTK}$ | $v_0$ |
|-------|-----------|-------|----------------|-------|
| 3 | 15 | $0.980 \pm 0.017$ | $0.889 \pm 0.068$ | $0.980 \pm 0.011$ |
|   | 30 | $0.924 \pm 0.110$ | $0.909 \pm 0.063$ | $0.985 \pm 0.006$ |
|   | 60 | $0.830 \pm 0.208$ | $0.946 \pm 0.043$ | $0.988 \pm 0.004$ |
|   | 90 | $0.856 \pm 0.220$ | $0.967 \pm 0.020$ | $0.986 \pm 0.006$ |
| 4 | 15 | $0.983 \pm 0.015$ | $0.925 \pm 0.059$ | $0.982 \pm 0.009$ |
|   | 30 | $0.964 \pm 0.036$ | $0.956 \pm 0.039$ | $0.985 \pm 0.007$ |
|   | 60 | $0.920 \pm 0.152$ | $0.955 \pm 0.036$ | $0.986 \pm 0.008$ |
|   | 90 | $0.974 \pm 0.026$ | $0.961 \pm 0.031$ | $0.984 \pm 0.009$ |
| 5 | 15 | $0.985 \pm 0.011$ | $0.921 \pm 0.049$ | $0.978 \pm 0.010$ |
|   | 30 | $0.969 \pm 0.023$ | $0.953 \pm 0.038$ | $0.983 \pm 0.008$ |
|   | 60 | $0.947 \pm 0.066$ | $0.966 \pm 0.033$ | $0.982 \pm 0.028$ |
|   | 90 | $0.959 \pm 0.037$ | $0.974 \pm 0.026$ | $0.982 \pm 0.010$ |

transformations. The results are shown in Table 3. Similarly, we evaluate our approximation of the principle eigenvector $v_0$, by looking at the absolute cosine distance between our approximation and the ground-truth.

Finally, in Section 3.3, we approximated the principal eigenvector $v_0$ of the NTK $K$ with the row mean $K1/1^\top 1$. The median cosine alignment between the row mean and the true $v_0$ was found to be 0.99995 across all epochs surveyed, across all models and datasets. The IQR is 0.00446. The strength of this approximation is a testament to the extreme spectral gap of the NTK, which itself is a consequence of NTK alignment.

# G    Additional Experimental Results

## G.1    Order Parameter Trajectories for Single Runs

This section contains additional illustrations of the order parameter trajectories, and the corresponding confidence region estimates, similar to the left side of Figure 3. The results are shown in Figure 19. Each model is a 5 layer deep, 128-unit wide SIREN network, trained with full-batch gradient-descent with a learning rate of 1e-3.

## G.2    Influence of Hyperparameters on Order Parameter Trajectories

In this section, we perform an ablation study to understand the impact of different hyperparameters on the order parameter trajectories. The baseline model is a 5-layer 128-unit wide SIREN with $\omega_0 = 60$. Figures 20-22 showcase the effect of depth. Figures 23-25 showcase the effect of the bandwidth parameter $\omega$.

When depth (and therefore model capacity) is decreased, we observe a corresponding increase in the validation error. In shallower models, the initial gradient confusion ($\min C_{NTK}$) is lower, delaying learning, and thus, the peak in the loss rate $\dot{L}_{eval}$. While the location of the phase transition changes, the trajectory of the order parameters shapes remain consistent, and exhibit less variance with increased depth. By contrast, there is dramatic change in the shape of the trajectories as we vary $\omega_0$. When $\omega_0$ is high, $\xi_{corr}$ starts very low, favouring interactions with immediate neighbours, leading to low overlap with the RBF. During training, the range broadens rapidly, causing $\mathrm{CKA}(K_X, K_{NTK})$ to grow sigmoidally at the critical point. Conversely, with low $\omega_0$, the range starts large but shrinks during training.

We additionally track the CKA between the NTK and a static RBF kernel with fixed bandwidth $K_X$, as described in 4.1. The evolution of this hyperparameter reflects the evolution of the correlation lengthscale $\xi_{corr}$. When this value is large (as it is when $\omega_0$ is small), the NTK has a broad diagonal,

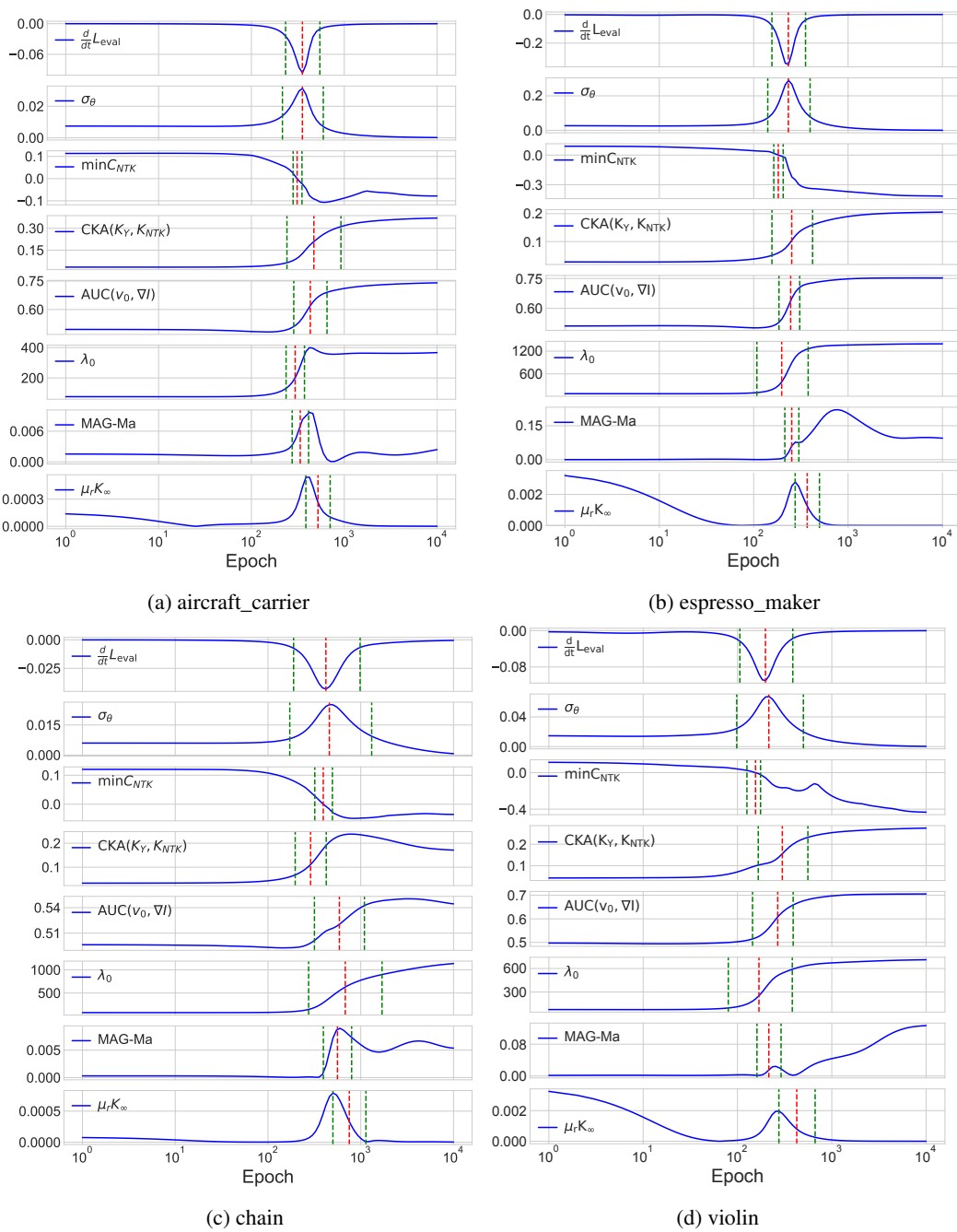

(a) aircraft_carrier

(b) espresso_maker

(c) chain

(d) violin

Figure 19: **Alignment of Order Parameters**. Order parameter evolution and critical points during training of a SIREN model. The red vertical lines denote the location of the critical points, and the green vertical lines denote confidence regions.

and thus overlaps well with the RBF. Over the course of training, $\xi_{corr}$ shrinks, and thus, so does $\mathrm{CKA}(K_X, K_{NTK})$.

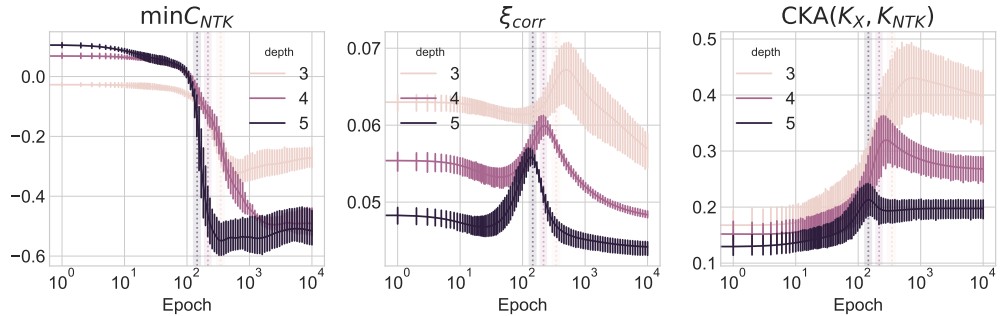

Figure 20: **Effect of depth on Critical Behaviour (Microphone)**: Average MSEs, in order of ascending depth: $2.561e^{-2} \pm 9.355e^{-5}$, $2.555e^{-2} \pm 8.970e^{-5}$, $2.572e^{-2} \pm 7.209e^{-5}$. Dashed vertical lines denote the location of the peak of the loss rate $\dot{L}_{\text{eval}}$, marking the phase transition.

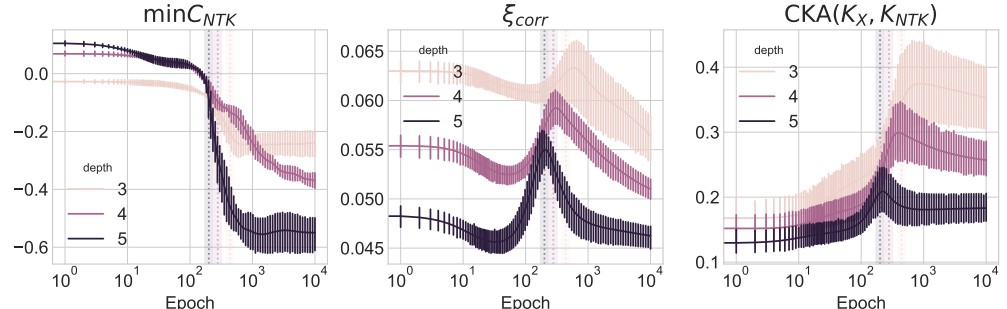

Figure 21: **Effect of depth on Critical Behaviour (Sax)**: Average MSEs, in order of ascending depth: $1.628e^{-2} \pm 1.312e^{-4}$, $1.513e^{-2} \pm 3.384e^{-5}$, $1.494e^{-2} \pm 6.605e^{-5}$. Dashed vertical lines denote the location of the peak of the loss rate $\dot{L}_{\text{eval}}$, marking the phase transition.

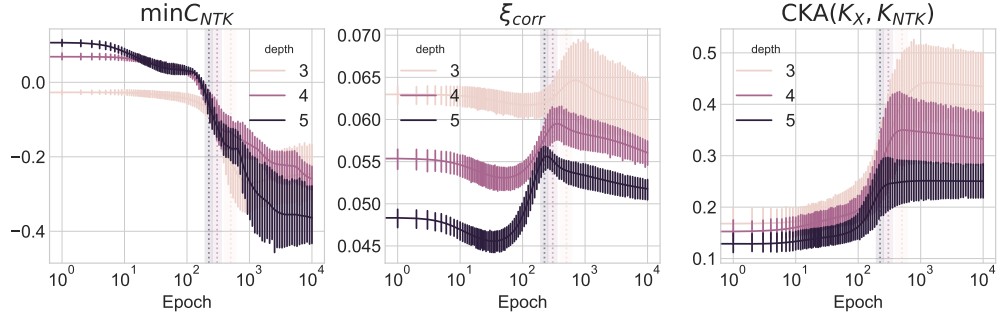

Figure 22: **Effect of depth on Critical Behaviour (Violin)**: Average MSEs, in order of ascending depth: $6.885e^{-3} \pm 1.677e^{-4}$, $5.930e^{-3} \pm 5.016e^{-5}$, $5.665e^{-3} \pm 3.640e^{-5}$. Dashed vertical lines denote the location of the peak of the loss rate $\dot{L}_{\text{eval}}$, marking the phase transition.

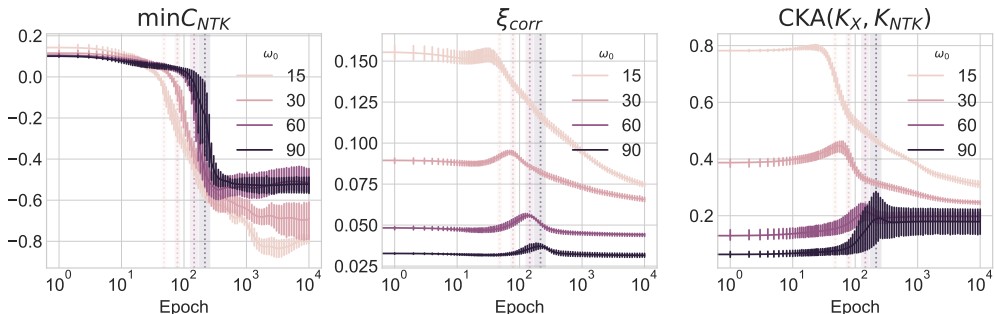

Figure 23: **Effect of $\omega_0$ on Critical Behaviour (Microphone)**: Average MSEs, in order of ascending $\omega_0$: $2.601e^{-2} \pm 1.804e^{-4}$, $2.566e^{-2} \pm 1.327e^{-4}$, $2.572e^{-2} \pm 7.209e^{-5}$, $2.807e^{-2} \pm 7.688e^{-4}$. Dashed vertical lines denote the location of the peak of the loss rate $\dot{L}_{\text{eval}}$, marking the phase transition.

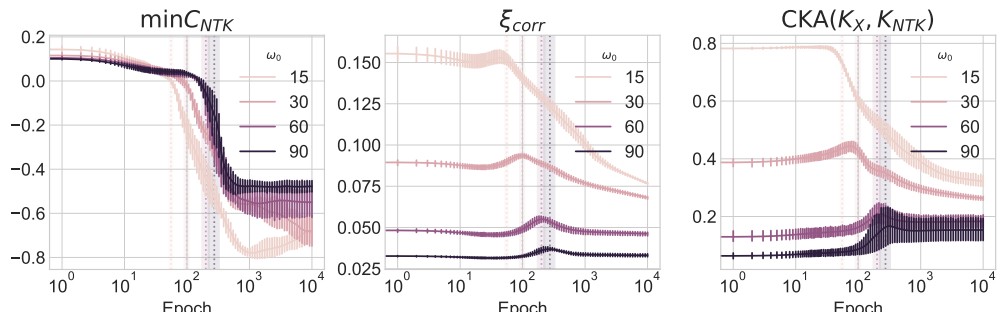

Figure 24: **Effect of $\omega_0$ on Critical Behaviour (Sax)**: Average MSEs, in order of ascending $\omega_0$: $1.680e^{-2} \pm 1.666e^{-4}$, $1.561e^{-2} \pm 6.552e^{-5}$, $1.494e^{-2} \pm 6.605e^{-5}$, $1.639e^{-2} \pm 3.938e^{-4}$. Dashed vertical lines denote the location of the peak of the loss rate $\dot{L}_{\text{eval}}$, marking the phase transition.

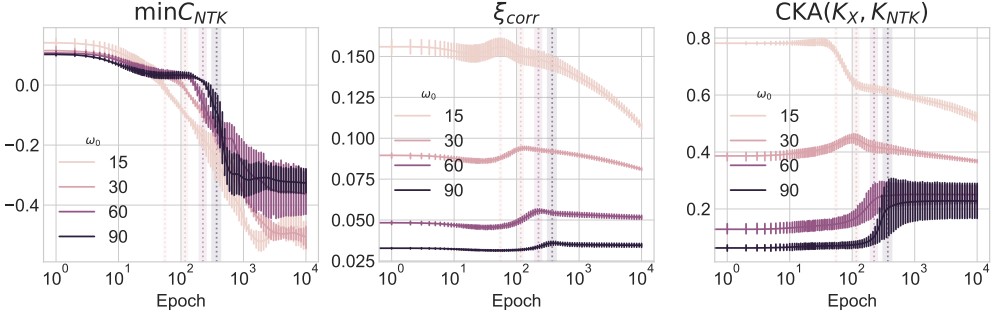

Figure 25: **Effect of $\omega_0$ on Critical Behaviour (Violin)**: Average MSEs, in order of ascending $\omega_0$: $7.223e^{-3} \pm 1.503e^{-4}$, $6.305e^{-3} \pm 3.139e^{-5}$, $5.665e^{-3} \pm 3.640e^{-5}$, $6.698e^{-3} \pm 3.359e^{-4}$. Dashed vertical lines denote the location of the peak of the loss rate $\dot{L}_{\text{eval}}$, marking the phase transition.

