# OpenReview forum: "A Closer Look at NTK Alignment: Linking Phase Transitions in Deep Image Regression"
_NeurIPS.cc/2025/Conference — NeurIPS 2025 poster_

### Official Review · Reviewer_TPRi · 2025-06-20

**Clarity:** 1
**Significance:** 2
**Originality:** 3
**Rating:** 4
**Confidence:** 3

**Summary:**

The paper studies the NTK alignment for image regression models a more complicated non-linear training dynamics problem. They find that the new observation of wave patterns simultaneous occurs with two other phase transitions that of the loss rate collapse and NTK alignment. Moreover, the presence of these phase transitions is both derived theoretically and illustrated in experiments. The evaluation is done by tracking the corresponding order parameters and visualizations of the NTK at several time steps.

**Questions:**

Does the onset of wavecrests not appear when the residuals are small? Therefore, they would directly coincide with the decaying loss.

Eq.(10) has a Laplacian operator and and a time derivative in it with respect to which variables are they? Given that previously theta seems to evolve with time and that the Laplacian is with respect to x it seems that it is not a real diffusion equation. Could the authors please clarify this?

What does the wavecrest effect imply in general? Is the effect desirable, neutral or bad?

In Figure 1a, what is the scaling of each plot?

**Ethical Concerns:**

["NO or VERY MINOR ethics concerns only"]

**Final Justification:**

The clarifications of the authors where helpful to understand the nature of their contributions better. This eases my initial concern with the structure of the paper. However, I believe it still can be improved and the authors have made concrete plans for doing so. To summarize their contribution is interesting and it is possible to follow their main storyline. Therefore, I have updated my score to borderline accept.

**Limitations:**

yes

**Quality:**

2

**Strengths And Weaknesses:**

Strengths
The motivation is clear and the study of NTK alignment for image regression is well founded.
The new observed phase transition in the residual is interesting and to the best of my knowledge novel.
The experimental validations supports the new derived observation of the wavecrests.

Weakness:
The main theoretical results are not in a theorem-proof format. This makes it harder to distinguish the contributions from previous known results. Moreover, it strongly decreases the readability and rigorousness of the manuscript.

The occurrence of the transitions at the same time-scale indicates that they are the same phase transition but different effects. Moreover, the significance of the wavecrest effect is unclear.

Furthermore, in the relate work there are only two recent papers cited from 2024 and 2023. For example relevant NTK literature might be:

Yu, Zixiong et al. “Divergence of Empirical Neural Tangent Kernel in Classification Problems.” ArXiv abs/2504.11130 (2025): n. pag.

Khalafi, S., Sihag, S. &amp; Ribeiro, A.. (2024). Neural Tangent Kernels Motivate Cross-Covariance Graphs in Neural Networks. <i>Proceedings of the 41st International Conference on Machine Learning</i>, in <i>Proceedings of Machine Learning Research</i> 235:23577-23621 Available from https://proceedings.mlr.press/v235/khalafi24a.html.

The work holds promise but in its current form falls short of the conference' standards. Improving the structure of the theoretical contributions is key.

Miscellaneous: typos and style

Table1: the word hyperparamters

Same line 82

line 179: 21 should be (21)

The hyperparameters mentioned in the text for Figure 4 mismatch the ones in the plot.

Almost all formulas are numbered this is unnecessary.

---

> ### Author Rebuttal · Authors · 2025-07-28
>
> We thank the reviewer for taking the time to read out paper and leaving thoughtful suggestions.
>
> ### Concerning Paper Organization
>
> We acknowledge your feedback regarding the lack of a ‘theorem-proof’ structure for the theoretical results, and recognize that it can help us distinguish our contributions.  Our approach is primarily phenomenological, focusing on deriving approximate closed-form expressions based on empirical observations and validating them experimentally. While we did not conceptualize these results as formal proofs, we did include derivations and detailed justifications in the appendix.
>
> We understand that presenting these results in a theorem-proof format may enhance clarity and rigor, and we would be happy to adopt this structure in a revision.  This seems like a minor structural change that could be accomplished without major re-structuring.  Unfortunately we cannot upload revisions during this rebuttal, but as some examples:
> - In Section 3.1, we would merge equations (6,7,8) into a Proposition block, as these are import ansatz/approximations.  Eq 10 would then be wrapped inside a theorem block.  The subsequent Proof, which would occupt the remainder of Section 3.1, would still be a high-level sketch due to constraints.  However the more rigorous derivation contained in Appendix A.2 could certainly be formatted using this structure.
> - The other important theorem block would wrap around our Cauchy Approximation in Eq 12, and a subsequent proof would be offered.  Equations 15, 20, and 21 would be labelled as Corollaries, since we derive them as consequences of the main theorem.
>
> In summary, we believe, for the camera-ready version, the theorem-proof structure could be adopted via (1) minor formatting adjustments and (2) minor shuffling within the sections.  The overall structure of the paper, the results, figures, and contributions, would be otherwise unchanged.
> ### Concerning the significance of wavecrests
>
> We understand that many of your concerns primarily focus on the wavecrest phenomenon.  Notably:
>
> > The occurrence of the transitions at the same time-scale indicates that they are the same phase transition but different effects. Moreover, the significance of the wavecrest effect is unclear...Does the onset of wavecrests not appear when the residuals are small? Therefore, they would directly coincide with the decaying loss.
>
> The reviewer is correct in noting that the co-occurence of the phase transitions is indeed suggestive of a shared underlying mechanism.  However, it is not quite so simple as saying that the wavecrests appear when the residuals are "small." While we cannot include figures in this rebuttal, if one were to plot $\mu_r$ (the mean residual) as a function of time for the experimental run in Figure 3 (left-hand side), you would see the residuals become "small" around $t=32$ epochs, though the phase transition for the loss rate (after which we see wavecrests) occurs near $t=324$ epochs.
>
> To understand why this happens, we must look at Eq 35 for the evolution of the residuals.  We obtain this equation when we insert our Gaussian Ansatz for the NTK (Eq 8) into the residual evolution equation (Eq 3).  Eq 35 shows a competition of two terms:
> 1. The "local term," corresponding to the interactions between grid points and their close neighbours (defined by a correlation lengthscale $\xi_{corr}$).
> 2. The "long range term," corresponding to the product of the mean residual, the local gradient magnitudes, and the asymptotic value of the Cosine NTK.
>
> The appearance of wavecrests is assosciated with the vanishing of this long range term: it swells close to the critical point as gradient magnitudes swell and then decays once more (right subfigure in the provided plot).  Stated another way, **the appearance of wavecrests is a visual hallmark of the dominance of local interactions in the evolution of the residuals.**
>
> This is significant because it justifies our focus on the local structure of the NTK when constructing (1) our approximation of the principal eigenvector (Eq 20), and (2) our approximations for the minimum value of the Cos NTK (Eq 21).
>
> To answer your other question:
>
> > Is the effect desirable, neutral or bad?
>
> Strictly speaking the effect is neither bad nor good.  In this paper we focus on the timing of the transition, and how it aligns with the timing of other phase transitions.  Phase transitions in GD dynamics shed light onto the effective capacity of a model: given two architectures with the same capacity on the same dataset, if one transitions earlier, it likely converges earlier, and that might be indicative of under-fitting (which we explore in Section 4.1).
> ### Concerning the Laplacian in Eq 10
>
> In response to your question:
>
> > Eq.(10) has a Laplacian operator and and a time derivative in it with respect to which variables are they? Given that previously theta seems to evolve with time and that the Laplacian is with respect to x it seems that it is not a real diffusion equation. Could the authors please clarify this?
>
> The Laplacian is with respect to the spatial coordinate $x$.  This is stated explicitly in Appendix A.2 but this is an easy notational matter to correct in the camera-ready version of Eq 10.
>
> While the reviewer is correct in stating that parameters $\theta$ are the dynamical variable, these dynamics are very high-dimensional and difficult to analyze.  However, the NTK provides an alternative perspective on gradient descent (GD) dynamics.  In particular, if we take a continuum limit in both the time (gradient flow) and the data, we end up with field equations for the evolution of the residuals.
>
> More explicitly, the residual 'field' is $r(x,t) = I(x) - f(x;\theta(t))$, where $I(x)$ is the image intensity at the coordinate $x$, and $f(x;\theta(t))$ denotes the output of the model at the point $x$ given the parameters $\theta$ at time $t$.  Under the NTK, the evolution of the residual field looks like (per Eq 3):
> $$
> \dot{r}(x,t) = -\int du\; r(x+u,t) K_{NTK}(x,x+u;\theta(t))
> $$
> In our work, we propose phenomenological approximations for the structure of $K_{NTK}$ based on experimental observation.  One such model is a Gaussian with spatially-varying correlation length (Eq 8), motivated by the rapid decay of correlation-functions with separation distance $||u||$.
>
> Because the NTK is now approximated as a Gaussian, the evolution equation above can be explicitly integrated.  To do this, we approximate $r(x+u,t)$ via a second-order Taylor expansion in the separation $u$ (Eq 9), which is an expansion in the spatial derivatives of $r(x+u)$.  The full details are in appendix A.2.  The Laplacian term emerges because it is the trace of the Hessian $\nabla_x^2 r$ in the Taylor expansion (Eq 19).
>
> In summary, because the NTK strongly favours local interactions, the first and second order spatial gradients of $r$ tell us a lot about how it evolves in time, because these describe the local structure of the residuals.  We think this can be easily clarified using the suggested revision we shared with Reviewer Myuw:
>
> *An exponentially-decaying NTK will suppress all contributions to $\dot{r}(x)$ from residuals $r(x+u)$ for which $||u|| \gg \xi(x)$.  When $\xi(x)$ is small, only the local neighbourhood of $r(x)$ influences its dynamics, justifying a Taylor expansion around the point $x$...*
>
> ### Concerning Related Works
>
> We thank the reviewer for directing us to the two recent NTK works.  While the work on the divergence of the NTK is rather distinct from our own, the work on Cross-Covariance Graphs further reinforces the role that NTK alignment has in understanding convergence in GD (indeed, the paper's theoretical contributions shows that the convergence of GD is positively correlated with their alignment metric).  However, in the referenced paper, the goal is to directly maximize alignment, whereas in our investigation, we study how, when, and why NTK alignment happens *spontaneously* in GD without direction supervision.  We will add a sentence to our Related Works section for the camera-ready version that highlights this distinction.  Something to the effect of (at the end of Section 6):
>
> *...contributing to the superior performance of DNNs over models based on infinite-width NTKs [26].  **Direct optimisation of alignment measures has even been suggested as one way to enhance the convergence of GD [cite the provided work and others].**  That said, theoretical investigation into spontaneous NTKA often focus on shallow networks...*

---

> > ### Comment · Reviewer_TPRi · 2025-08-04
> > **Response to rebuttal**
> >
> > I appreciate the efforts of the authors for addressing most of my concerns.
> > However I still have some remaining concern.
> > 1. My concern is still with the formatting of the exposition.
> > It would be appreciated if the authors at least formalize the first statement of Section 3.1 in a theorem and post it as response.
> > 2. I would like to know what the scaling of Figure 1a is? The scaling is important for the interpretation of the main phenomenon.    A linear or log scaling could lead to very different figures for the residual. Can the authors explain which scaling it is and why?
> >
> > Given the clarification of the nature of the paper and reading the other reviews I will increase my score accordingly.
> > This is under the assumption that the authors provide one theorem statement.

---

> ### Author Response · Authors · 2025-08-07
> **Response to Reviewer TPRi**
>
> Before we share the formalisation of the contribution in Section 3.1, we wanted to quickly address your other concern.  Regarding the scaling of Fig 1a, it is a log scale, from $10^0$ (yellow) to $10^{-8}$ (dark purple).  We will include a colorbar in the camera ready version.
>
> As for our proposed formalisation, see below:
>
> ---
>
> Ansatz 1: **Gaussian Approximation of NTK**.  Ignoring anisotropic effects, the NTK may be approximated as a Gaussian kernel with spatially-varying amplitude $||\nabla_{\theta} f(x)||^2$, bandwidth $\xi(x)$, and asymptotic value $||\nabla_{\theta} f||^2 c_{\infty}(x)$:
>
> $K_{Gauss}(x,x+u) \equiv ||\nabla_{\theta}f(x)||^2\exp(-||u||^2/\xi^2(x)) + ||\nabla_{\theta}f(x)||^2 c_{\infty}(x)$
>
> Theorem 1: **Diffusive Evolution of the Residuals** Consider the residuals $r(x,t) = I(x) - f(x;\theta(t))$ for a DNN $f_{\theta}$ trained to reconstruct a signal $I(x)$ using gradient flow in the continuum limit, where $I, f_{\theta} \in C^{\infty}$.  Assume the NTK $K_{NTK} \approx K_{Gauss}$.  Let the mean residual be denoted by $\mu_r \equiv \mathbb{E}[r]$, and let $\delta r_{\infty}(x) \equiv \mu_r ||\nabla_{\theta}f(x)||^2 c_{\infty}(x)$ be the contribution to the residual evolution at large distances $||u||  \gg \xi(x)$.  Then, as $\delta r_{\infty}(x) \to 0$, the residuals evolve under the following diffusion equation:
>
> $\frac{d}{dt}r(x,t) \approx -2\pi\xi^2(x) ||\nabla_{\theta}f(x)||^2 r(x,t) - \pi \xi^4(x) ||\nabla_{\theta}f(x)||^2 \Delta_x^2r(x,t)$

---

### Official Review · Reviewer_RheY · 2025-06-29

**Clarity:** 3
**Significance:** 3
**Originality:** 3
**Rating:** 4
**Confidence:** 2

**Summary:**

The paper proposes a unifying, Neural-Tangent-Kernel (NTK)-based framework for constructing order parameters that make the dynamics of deep image-regression models (SIRENs) more interpretable. Analytically, it derives a local Gaussian/Cauchy approximation of the NTK, then shows how three apparently distinct phase transitions during training — (i) emergence of diffusion-like wave-crests in residuals, (ii) collapse of the loss-rate slope, and (iii) sudden NTK alignment with image edges. Empirically, the study demonstrates that the critical points for these transitions cluster in time and show that their relation over the network depth and the sinusoidal bandwidth hyper-parameter $\omega_0$.

**Questions:**

- What happens If each layer adopts its own bandwidth $\omega_0$ ?  The corresponding results will be dramatically changed ?
- Can the Gaussian + Cauchy NTK approach generalize to 3-D radiance-field tasks ?

**Ethical Concerns:**

["NO or VERY MINOR ethics concerns only"]

**Final Justification:**

The rebuttal clearly explains why per-layer bandwidth mainly matters in the first layer and gives a reasonable path for extending the NTK analysis to 3-D radiance fields; this resolves my main conceptual concerns. The theory is solid and the clarifications are credible, but empirical support remains thin. On balance, I recommend borderline accept.

**Limitations:**

Yes

**Quality:**

3

**Strengths And Weaknesses:**

##  Strengths
- Rigorous, closed-form approximation of the local SIREN NTK, leading to explicit expressions for correlation length, minimal cosine NTK, and principal eigenvector.
- First report of diffusion-like wave-crests in INR residuals, linked analytically to short-range NTK structure.
- Clear practical insight: deeper networks or larger $\omega_0$ postpone critical points and improve super-resolution accuracy, whereas small $\omega_0$ can cause premature convergence.

## Weakness
- Scope is restricted to full-batch gradient descent on 2-D grayscale super-resolution.
- Computing NTK spectra for 256×256 evaluation grids is expensive; runtime and memory requirements are not quantified.

---

> ### Author Rebuttal · Authors · 2025-07-28
>
> We thank the reviewer for taking the time to read out paper and leaving thoughtful suggestions.  To address your questions:
>
> > What happens If each layer adopts its own bandwidth ω0 ? The corresponding results will be dramatically changed ?
>
> This is a subtle question.  As [30] (the Fourier Features paper) shows, MLPs have specific difficulty learning high-frequency functions from low-dimensional inputs. The authors specifically highlight how the NTK is poorly conditioned in low-dimensional spaces, and how fourier feature mappings can help alleviate this.  The intermediate layers, by contrast, are mapping between high dimensional latent spaces, for which the spectral bias isn't so damning.  The SIREN architecture (used in this paper) can be seen as a learnable fourier mapping, whose conditioning is controlled by the bandwidth $\omega_0$.  As such, we expect that tuning the $\omega$ of the intermediate layers would have significantly less impact than tuning the first layer bandwidth.
>
> > Can the Gaussian + Cauchy NTK approach generalize to 3-D radiance-field tasks ?
>
> We expect that our results should generalise, all else being equal.
>
> 3D radiance field tasks take in low-dimensional inputs (here, 3D position + view direction) and predict RGB$\sigma$ values through mean-squared error (MSE) minimization.  This setup is very similar to the grayscale INR case considered in this paper, with two notable differences:
> - The outputs for 3D radiance tasks are higher-dimensional, but the MSE criterion is still used.  Thus, via the chain rule, little changes in Eq 3 save that the matrix equation becomes a matrix equation between flattened matrices.
> - The inputs for the 3D-radiance task are higher-dimensional.  In this case, the most crucial question has to do with the correlation lengthscale.  If we still see rapid decay in $K_{NTK}(x,x+u)$ for separation distance $||u||$, then all of our derivations should hold, since they are principally based on Taylor expansions justified by the locality of the NTK.  To that end, we expect that our results should generalize.  Refrence [31] shows that, at initialization in the large width limit, the SIREN NTK decays like a sinc function.  However, experiments will need to confirm whether this result holds across training as it does for our case.

---

> > ### Comment · Reviewer_RheY · 2025-08-09
> >
> > Thank you for addressing my concerns.

---

### Official Review · Reviewer_Myuw · 2025-07-01

**Clarity:** 3
**Significance:** 3
**Originality:** 3
**Rating:** 4
**Confidence:** 3

**Summary:**

In this article, the authors aim to develop a unified framework that uses order parameters to characterize different learning phases by leveraging the NTK framework. Specifically, the authors focus on deep image regression with deep sinusoidal representation networks (SIRENs) to address a more complex use case than existing works, which typically employ two-layer or deep linear models. By using the NTK formalism, the authors identify 3 order parameters that delineate three learning phases:
 - Diffusion-like behavior of the network residuals,
 - Alignment of the network NTK,
 - Loss-rate collapse.

Subsequently, through empirical analysis, they demonstrate that these order parameters undergo change almost simultaneously over training, and thus argue for a shared underlying mechanism controlling the learning phases described by them.

**Questions:**

- Is the fact that the NT-kernel measure in this work is defined across different spatial points of the same image (rather than different examples/images, as in a typical setting) the cause for it to have negative eigenvalues?
- Which of the order parameters from Figure 3 would correspond to depicting the onset of diffusion?
- How does AUC best capture alignment and “edge affinity,” in particular? Furthermore, why is this metric chosen over other alternatives?
- While the authors note that the Gaussian NTK approximation holds “anytime away from a phase transition,” its applicability to generative regression tasks is claimed but not supported by empirical or theoretical evidence.
- The finding that the NTK “is conveying information about image edges” raises the question: why edges specifically, and how would the framework adapt to other salient structures or modalities?

**Ethical Concerns:**

["NO or VERY MINOR ethics concerns only"]

**Limitations:**

See Weaknesses & Questions.

**Paper Formatting Concerns:**

N.A.

**Quality:**

3

**Strengths And Weaknesses:**

- $\textbf{Strengths}$:
  - The authors consider the non-trivial case of SIREN models, moving beyond simpler 2-layer or DLNs considered in prior work, in their analysis
  - Under an isotropy assumption (on the data), the authors demonstrate that a diffusion equation describes the evolution of the residual
  - Under practical conditions, where the above assumption does not apply, the authors derive a closed-form expression for the correlation length, which acts as an order parameter to describe the onset of diffusion.
  - Similarly, the authors provide closed-form expressions for the other two order parameters, namely the top-eigenvector of the NTK and the minimum value of the related cosine-NTK
  - The authors observe that the critical epoch occurrence can be preponed by:
    - By increasing network depth
    - By decreasing $\omega_0$
  - The authors observe that:
    - Lowering $\omega_0$ causes gradient-energy hotspots to concentrate sharply on true image edges
    - $\log\|\nabla_θf(x)\|^2$ and $v_0(x)$ show high spatial correlation with $\|\nabla_x I(x)\|$, confirming that edge pixels drive NTK alignment
    - Deeper models maintain richer, more distributed gradient features before focusing, enabling better utilization of model capacity.

- $\textbf{Weaknesses}$:
  - Generally speaking, the clarity, as well as notational inconsistencies, of the paper can be improved.
  - Key approximations are stated, but intermediate algebra is too often relegated to appendices. Hence, the reader can struggle to bridge the gap from Eq. (6) / (7) ⟶ (15).
  - Reliance on a second-order Taylor expansion in neighborhood distance presumes small perturbations; the boundary of this approximation’s validity vis-à-vis real image data is not discussed.
  - MAG-Ma is introduced without sufficient theoretical motivation or linkage to prior stationarity-breakdown measures, leaving its relevance and novelty in question.
  - It is unclear what additional information Eq. 12 provides about the loss gradient beyond what is already encapsulated by the order parameter $\sigma(\theta)$.
  - Basing full-batch gradient-flow experiments on only 30 samples raises questions about the statistical significance and the generalizability of the conclusions drawn from the results.
  - The caption for Figure 3 suggests that order-parameter trajectories were studied on a single image. If true, it precludes assessment of variance or significance across a representative dataset.
  - A fixed hidden-layer width of 256 may not suffice to enter the NTK regime; without width-scaling experiments, it’s unclear whether the empirical setup aligns with the asymptotic NTK assumptions.
  - It is unclear what the alignment between the theoretical predictions in this article and that of a practical training regime (away from gradient flow dynamics, e.g., SGD) would be.

  - $\textbf{Notation}$:
    - C_{NTK}: Previous literature (e.g., Arora et al., 2019) uses CNTK to refer to Convolutional NTKs. The notation used here, while slightly different in appearance, can be confused with that.
      - On line 131, you again use Cos NTK
    - Line 116:  ξ should be ξcorr
    - Line 179:  minimum of 21 should be minimum of Eq. 12
    - Line 187: The reference for Centred Kernel Alignment is not provided

---

> ### Author Rebuttal · Authors · 2025-07-28
>
> We sincerely thank the reviewer for their thoughtful feedback and the specificity of their questions.
> ### Response to Weaknesses
>
> Before we answer the reviewer's questions, we would like to address some of the concerns raised re weaknesses.
>
> > The caption for Figure 3 suggests that order-parameter trajectories were studied on a single image. If true, it precludes assessment of variance or significance across a representative dataset.
>
> Each order parameter trajectory is indeed obtained from a single training run ie a pairing of an image (the dataset) and a model architecture.  To clarify, it is not possible to obtain a single order parameter trajectory from multiple images because our models are trained on only one image at a time (this is the setup commonly used in constructing implicit neural representations of images).
>
> That said, there are 3600 runs total, corresponding to the experimental sweep described in Section 4 and more detail in Appendix B.2.  However, direct comparison between trajectories is difficult: the scale of the order parameters and the timing of the phase transitions varies quite significantly as a function of model and dataset.  An example of this variation can be seen in Figure 13, where multiple order parameter trajectories are shown together in the same plot.
>
> However, **this does not preclude** examination across datasets/architectures:
> 1. Within a given run, we can look for the intersection rate of the confidence regions for the different phase transitions.  The result of this is the heatmap shown in Figure 3, whereby we see the remarkable coincidence between the different phase transitions.
> 2. When we groupby hyperparameters such as depth and $\omega_0$, this variation becomes more tame, and it becomes possible to compare statistics between runs.  An example of this analysis is found in Section 4.1, with the expectation and variance across runs shown in Table 1.
>
> > It is unclear what additional information Eq. 12 provides about the loss gradient beyond what is already encapsulated by the order parameter σ(θ).
>
> We politely ask the reviewer to clarify this question.  $\sigma(\theta)$ describes the loss gradient variance and Eq 12 is the point-wise local structure of the NTK.  The latter is a significantly richer object - is the reviewer perhaps referring to another equation?
>
> > Reliance on a second-order Taylor expansion in neighborhood distance presumes small perturbations; the boundary of this approximation’s validity vis-à-vis real image data is not discussed.
>
> We thank the reviewer for highlighting this point.  It was discussed in the paper, but we agree it could me made more explicit.
>
> As shown in Equation 8 and discussed throughout Section 3.1, the relevant lengthscale to consider when performing the Taylor expansion is the correlation lengthscale.  Typical values for the correlation lengthscale can be seen in Figure 2, Table 1, and Figure 17 of the Appendices.
>
> We believe the best place to make this clearer is in the snippet before Eq 9.  In particular, the camera-ready version will say something to the effect of:
>
> *An exponentially-decaying NTK will suppress all contributions to $\dot{r}(x)$ from residuals $r(x+u)$ for which $||u|| \gg \xi(x)$.  When $\xi(x)$ is small, only the local neighbourhood of $r(x)$ influences its dynamics, justifying a Taylor expansion around the point $x$...*
>
> > MAG-Ma is introduced without sufficient theoretical motivation or linkage to prior stationarity-breakdown measures, leaving its relevance and novelty in question.
>
> We are unaware of any studies that track the dynamical breakdown of stationarity in the NTK, and would be grateful if the reviewer could point us towards a reference.
>
> Our theoretical justification for MAG-Ma is alluded to at the end of Section 3.3, and described in much greater detail in Appendix E.2.  In stat physics, order parameters are often derived as a measure of symmetry violation, with the phase transition heralding a symmetry breaking.  In this work, we showed how various order parameters for SIREN training dynamics could approximated in terms of the local structure of the NTK.  This local structure is in turn determined by the spatial gradients of the parameter magnitudes, ie. $\nabla_x||\nabla_{\theta}f||^2$.  Hence, it makes sense to consider translation symmetry (ie stationarity) of the NTK to be the symmetry that is broken at the phase transition.  MAG-Ma is derived to measure this violation in symmetry explicitly.  We include it for the sake of completeness, and indeed find that there is a sudden shift in its value near other phase transitions.
>
> We do not consider this conclusive evidence of a shared underlying mechanism, but it is interesting, especially in connection with edge/corner detectors in computer vision.  Here, translation symmetry plays a crucial role in the identification of image features by studying auto-correlation functions.  This connection is explored more in Section E.1.
>
> > A fixed hidden-layer width of 256 may not suffice to enter the NTK regime; without width-scaling experiments, it’s unclear whether the empirical setup aligns with the asymptotic NTK assumptions.
>
> This paper does not focus on the asymptotic NTK regime (i.e., the lazy-learning regime of infinitely wide networks), but rather on the practical (ie finite-width) settings where NTK alignment is commonly observed.  As discussed in the Introduction and Section 6 (Related Work), the asymptotic NTK regime is a poor approximation for studying SIREN models and, more broadly, for understanding real-world neural networks, where NTK alignment evolves dynamically instead of remaining fixed.  Furthermore, prior work (e.g., [15,16,17,18,19,20]) demonstrates that this dynamic NTK evolution is a key factor in the superior performance of real-world DNNs compared to their infinite-width counterparts, as it reflects feature learning rather than mere kernel behavior. Our investigation specifically aims to (1) characterize the timing of this dynamic alignment by framing it as a phase transition and (2) explore the patterns learned in implicit neural representation tasks.
>
> >  It is unclear what the alignment between the theoretical predictions in this article and that of a practical training regime (away from gradient flow dynamics, e.g., SGD) would be.
>
> This is a limitation common to other works on NTK alignment (ex [21,22,24,25,26]).  We agree that it is worth investigating, but it is out of scope for the present work.  We retained the same setup from previous works but generalized the analysis to more complex models.
>
> ### Response to Questions
>
> > Is the fact that the NT-kernel measure in this work is defined across different spatial points of the same image (rather than different examples/images, as in a typical setting) the cause for it to have negative eigenvalues?
>
> The NTK is PSD by construction, hence there are never negative eigenvalues.
>
> The source of this misunderstanding is likely the beginning of section 3.2, where we described the Gaussian approx as positive definite ie the $K(x_i, x_i) > 0$ regardless of the points $(x_i, x_j)$ studied.  Empirically, however, gradients become unaligned, so that we may have points where $K(x_i, x_j) < 0$.  This is easy to correct for the camera ready revision.
>
> > Which of the order parameters from Figure 3 would correspond to depicting the onset of diffusion?
>
> The relevant order parameter is $K_{\infty}$, as defined on line 118. When this decays to zero, it signifies the decay of long-range interactions, and thus the dominance of local interactions.  We show that such interactions can be modelled via a diffusion equation.  Trajectories can be seen on the bottom of Figure 3.
>
> > How does AUC best capture alignment and “edge affinity,” in particular? Furthermore, why is this metric chosen over other alternatives?
>
> Our reasoning is summarized briefly in Section 3.3, insofar as it is insensitive to monotonic transformations of the principal eigenvector $|v_0|$.
>
> In more detail, this insensitivity is important because we want to study the alignment across different models and datasets (which all have very different distributions for the gradient magnitudes $||\nabla_x I||$).  We settled on using a canny edge detector as providing a "ground truth" definition of 'edginess' of the point $x$ as a binary label.  Treating $|v_0(x)|$ as a score for $x$, the AUC enables us to measure the utility of $|v_0(x)|$ in predicting edge membership (without needing to choose a threshold).
>
> > While the authors note that the Gaussian NTK approximation holds “anytime away from a phase transition,” its applicability to generative regression tasks is claimed but not supported by empirical or theoretical evidence.
>
> We politely ask the reviewer for clarification on this question.  We never claimed to study generative regression tasks, only supervised learning of INRs.
>
> > The finding that the NTK “is conveying information about image edges” raises the question: why edges specifically, and how would the framework adapt to other salient structures or modalities?
>
> We investigate this question at the end of Section 3.3, and in Appendix E.1.  In short, there are similarities between the NTK and the auto-correlation structure tensors used in computer vision for edge/corner detection.  Such detectors are based on the idea that edges/corners are more "complex" than flat regions, which is formalized through violations of translation symmetry/local variance.
>
> In DNN learning, higher local variance may cause DNNs to treat edges as "noisy datapoints."  Hence gradient energy accumulates at the points, which (via Eq 20), increases the prominence of such points in the principal eigenvector $v_0$.  We expect that other salient structures or modalities for other tasks should show similar behaviour: if a point has higher local-variance, the gradient magnitudes will diverge from its neighbours, and thus become more prominent in $v_0$.

---

### Official Review · Reviewer_qFsx · 2025-07-02

**Clarity:** 3
**Significance:** 3
**Originality:** 3
**Rating:** 4
**Confidence:** 1

**Summary:**

This paper theoretically establishes connections between the onset of NTKA and other dynamical phase transitions by deriving novel approximations for the local structure of the SIREN NTK and identify a novel learning phase in deep image regression, characterized by the appearance of diffusion-like wavecrests in the residuals. Experiments demonstrate the critical points for these different phase transitions cluster in time.

**Questions:**

I am not familiar with this area and topic, the only question I can raise can be seen in above weakness. Authors need to answer them.

**Ethical Concerns:**

["NO or VERY MINOR ethics concerns only"]

**Final Justification:**

I am not familiar with this area and topic，consider other reviewers' opnions, I would like to keep the score

**Limitations:**

Yes

**Quality:**

3

**Strengths And Weaknesses:**

Strengths:

1.This paper is the first to observe the emergence of diffusion-like wavecrests in residuals during SIREN training, theoretically linked to NTK, filling a gap in visualizing learning dynamics.

2.It empirically investigate the impact of image complexity and SIREN hyperparameters on the occurrence and timing of phase transitions,, offering practical guidance for SIREN design.

3.Sufficient experiments prove the validity of this theory.

Weakness:

1.The paper discusses SIREN model in the regression task of image reconstruction. I wonder if it can be used in other models or tasks?

2.The author doesn't seem to give the result of computational efficiency. It is suggested giving the result of computational efficiency of this method since computing eigenvectors requires high-dimensional operations.

---

> ### Author Rebuttal · Authors · 2025-07-28
>
> We'd like to thank the reviewer for taking the time to read and review our work.  To answer your questions:
>
> > The paper discusses SIREN model in the regression task of image reconstruction. I wonder if it can be used in other models or tasks?
>
> We believe that it can, with some modifications.  Rather than considering the residuals $r(x;\theta) = I(x) - f(x;\theta)$, we would consider the generalised residual $r(x;\theta) = \frac{\partial L}{\partial f(x;\theta)}$ Then, per the chain rule, we have:
>
> $\dot{r}(x;\theta) = \nabla_{\theta} r \cdot \dot{\theta} = \bigg(\frac{\partial^2 L}{\partial f(x)^2}\bigg)\nabla_{\theta}f(x)\cdot \dot{\theta}$
>
> The gradient flow dynamics are:
>
> $\dot{\theta} = -\nabla_{\theta}L = -\int dx^{\prime} r(x^{\prime};\theta) \nabla_{\theta}f(x^{\prime};\theta)$
>
> Inserting the above into the residual evolution equation yields:
>
> $\dot{r}(x;\theta) = -\bigg(\frac{\partial^2 L}{\partial f(x)^2}\bigg)\int dx^{\prime} r(x^{\prime};\theta) \nabla_{\theta}f(x^{\prime};\theta)\cdot \nabla_{\theta}f(x;\theta) = \bigg(\frac{\partial^2 L}{\partial f(x)^2}\bigg)\int dx^{\prime} r(x^{\prime};\theta) K_{NTK}(x,x^{\prime};\theta)$
>
>
>
> Hence, the NTK still emerges, although the residual equation is a little more complicated:
> - $r$ is now potentially a vector-valued function.
> - There is an additional, scalar coefficient that multiplies the NTK.
>
> Neither of these necessarily pose a problem for our approach.  The biggest question is whether the NTK $K_{NTK}$ experiences the same rapid decay with spatial separation.  At initialization, in the large width limit, the SIREN NTK decays like a sinc function [31].  If the decay remains sharp throughout training, then our derivation of the diffusion equation is largely unchanged: we make the Gaussian approximation, and perform a second-order Taylor expansion to obtain $r(x^{\prime})$ from $r(x)$ and its spatial gradients.  The Gaussian integration is unaltered, with the net result that the RHS of equation (10) for the diffusion equation is simply multiplied by $\bigg(\frac{\partial^2 L}{\partial f(x)^2}\bigg)$.
>
> In summary, there are reasons to believe that our results would generalize beyond the INR problem considered.  However, we would like to emphasise that, while this is an interesting future direction, such an investigation is not necessary to support the core results presented in this paper.
>
> > The author doesn't seem to give the result of computational efficiency. It is suggested giving the result of computational efficiency of this method since computing eigenvectors requires high-dimensional operations.
>
> To compute the eigenvectors of the NTK, we use a pytorch implementation of the Randomized SVD.  We described this in Section B.2 of the supplementary materials.  We can be more explicit by mentioning the method in the main body of the experimental section (Section 4) in the camera ready version.  That said, since the focus of our work lies in the analysis of the NTK rather than the computational method itself, we did not explicitly report detailed runtime performance.
>
> If the concern is whether or not our technique can be adapted to study larger datasets, the answer is yes.  Randomized SVD approximates the singular value decomposition by projecting the matrix onto a lower-dimensional subspace using random sampling, significantly reducing the computational complexity compared to exact SVD methods.  It is commonly used for dimensionality reduction of high-dimensional datasets with millions or even billions of rows, such as those that emerge in genomics studies or recommender systems.

---

> > ### Comment · Reviewer_qFsx · 2025-08-06
> >
> > I am not familiar with this area and topic，consider other reviewers' opnions, I would like to keep the score

---

### Decision · Program_Chairs · 2025-09-17

**Decision:**

Accept (poster)

**Comment:**

The paper presents  theoretical and empirical analysis of training dynamics in SIRENs. Its novel observation of wave patterns in residuals and the connection to NTK alignment and loss rate collapse offers interesting insight. The authors' rigorous rebuttals effectively addressed reviewer concerns regarding scope, methodology, and theoretical structure, solidifying the paper's contributions and making it a candidate for acceptance.